# 4D printed hydrogel scaffold with swelling-stiffening properties and programmable deformation for minimally invasive implantation

Bo Liu[1,2], Hui Li[1], Fengzhen Meng[1], Ziyang Xu[3], Liuzhi Hao[1,4], Yuan Yao[3], Hao Zhu[1], Chenmin Wang[1], Jun Wu[5], Shaoquan Bian[1], Willima W. Lu [1,6], Wenguang Liu [3] ✉, Haobo Pan [1] ✉ & Xiaoli Zhao [1,4] ✉

The power of three-dimensional printing in designing personalized scaffolds with precise dimensions and properties is well-known. However, minimally invasive implantation of complex scaffolds is still challenging. Here, we develop amphiphilic dynamic thermoset polyurethanes catering for multi-material four-dimensional printing to fabricate supportive scaffolds with body temperature-triggered shape memory and water-triggered programmable deformation. Shape memory effect enables the two-dimensional printed pattern to be fixed into temporary one-dimensional shape, facilitating trans-catheter delivery. Upon implantation, the body temperature triggers shape recovery of the one-dimensional shape to its original two-dimensional pattern. After swelling, the hydrated pattern undergoes programmable morphing into the desired three-dimensional structure because of swelling mismatch. The structure exhibits unusual soft-to-stiff transition due to the water-driven microphase separation formed between hydrophilic and hydrophobic chain segments. The integration of shape memory, programmable deformability, and swelling-stiffening properties makes the developed dynamic thermoset polyurethanes promising supportive void-filling scaffold materials for minimally invasive implantation.

Three-dimensional (3D) printing is extensively used in the biomedical field because it can construct complex biological structures with high precision and speed[1–3]. Customized hydrogel scaffolds with intricate architectures have shown promising results in biomedical applications, including medical devices, drug delivery carriers, and biological engineering[4–6]. However, the insertion of bulky 3D printed scaffolds often entails exposing the operation site to unavoidable injury. Given the growing preference for minimally invasive surgery in

---

[1]Research Center for Human Tissue and Organs Degeneration, Institute of Biomedicine and Biotechnology, Shenzhen Institute of Advanced Technology, Chinese Academy of Sciences, Shenzhen 518055, China. [2]Center for Health Science and Engineering, School of Health Sciences and Biomedical Engineering, Hebei University of Technology, Tianjin 300131, China. [3]School of Materials Science and Engineering, Tianjin Key Laboratory of Composite and Functional Materials, Tianjin University, Tianjin 300350, China. [4]University of Chinese Academy of Sciences, Beijing 100049, China. [5]Shenzhen Key Laboratory for Innovative Technology in Orthopaedic Trauma, Department of Orthopaedics and Traumatology, The University of Hong Kong-Shenzhen Hospital, Shenzhen 518048, China. [6]Department of Orthopaedics and Traumatology, Li Ka Shing Faculty of Medicine, The University of Hong Kong, Hong Kong 999077, China. ✉e-mail: wgliu@tju.edu.cn; hb.pan@siat.ac.cn; zhao.xl@siat.ac.cn

clinical practice[7,8], the ability to deliver 3D printed scaffolds in a compact state and autonomously morph in vivo to fit the implantation site would be advantageous.

Four-dimensional (4D) printing, introduced by Skylar Tibbits in 2013, involves using stimulus-responsive materials in conjunction with 3D printing technology[9]. The fourth dimension of 4D printing enables the printed scaffolds to undergo programmable deformations in response to specific stimuli, offering alternative possibilities for minimally invasive surgery[10,11]. In recent years, smart hydrogel inks have been developed to create 4D printed scaffolds capable of programmable deformations[12–14]. Currently, two main strategies are employed to achieve minimally invasive implantation. One approach is to compress and fix the printed 3D scaffold into a two-dimensional (2D) sheet outside the body, delivering it to the defect, and then the 2D sheet deforms in response to stimuli to return to its 3D structure for filling the targeted lesion[14,15]. The other method involves implanting the printed 2D sheet directly into the body, where it subsequently curls or folds into a 3D structure along a pre-determined path under stimulation[16–18]. However, the limited effect of single-dimensional deformation from 2D to 3D in minimizing trauma calls for further improvements. Trauma can be reduced by delivering scaffolds in a one-dimensional (1D) shape.

Shape memory polymers can exhibit temporary 1D shapes smaller than the catheter channel size under external forces, resulting in excellent capabilities for minimally invasive delivery. Leng et al. demonstrated a 4D printed occluder frame, which could be programmed into a temporary 1D linear shape based on temperature response, and recover its 3D double-disc shape under an alternating magnetic field[19]. However, external stimuli such as magnetic fields and high temperatures may impair tissue and organ function[20]. Endogenous stimuli, such as body temperature and water, are more convenient and biologically friendly options for further biomedical applications[21–23]. Nevertheless, printed hydrogel scaffolds with mechanical support and multidimensional morphing (1D to 3D) in response to mild endogenous stimuli have not been reported yet.

Multi-material 4D printing represents an emerging programmable manufacturing technique, enabling the construction of swelling mismatch structures that undergo programmable morphing in response to water, thus broadening the range of print materials for constructing hydrogel scaffolds[16,24,25]. The integration of shape memory polymer and multi-material 4D printing offers viable ideas for multidimensional morphing under the mild internal environment. However, scaffolds printed with conventional polymers inevitably experience mechanical loss during swelling-induced programmable deformation due to the plasticizing effect of water[26]. This phenomenon hinders the biomedical application of hydrogel scaffolds in high-humidity environments, where flexible scaffolds are desired for minimally invasive delivery while maintaining rigidity for mechanical support after implantation[27]. An alternative hydrogel with soft-stiff transition is needed to balance the deformation during transplantation and mechanical support at the targeted site. Recently, some amphiphilic polymers achieving water-induced stiffening based on water-driven phase separation have been reported[28–30], providing a promising idea for constructing swelling-stiffening hydrogel scaffolds.

Herein, we report an approach to deliver scaffolds in a minimally invasive manner through multidimensional morphing (1D to 3D) by the development of amphiphilic dynamic thermoset polyurethane (DTPU). Thermally reversible dynamic covalent bonds were introduced to impart viscoelastic rheological behavior for extrusion printing and enhance interlayer adhesion between the constituent layers. Through multi-material 4D printing based on fused deposition modeling (FDM), the DTPU can be printed into 2D scaffolds with body temperature-triggered shape memory and water-triggered programmable deformation (Fig. 1a). The printed 2D flat laminated patterns consist of a high-swelling DTPU as the active layer and a low-swelling

DTPU as the passive layer. Relying on the temperature-sensitive shape memory provided by the poly(ε-caprolactone) (PCL), the dry 2D pattern can be fixed into a temporary 1D roll-up shape. After immersion in 37 °C water, the 1D roll recovers to the initial 2D pattern and absorbs water to achieve programmed deformation, finally forming a 3D hydrogel scaffold. Simultaneously, swelling triggers segmental rearrangement, forming a phase separation structure and realizing a soft-to-stiff transition. Overall, the development of DTPU with this combination of features has demonstrated the feasibility of 4D printing scaffolds for minimally invasive implantation (Fig. 1b).

## Results

### Design and characterization of DTPU

To achieve DTPU with a combination of the aforementioned features, we selected PCL-triol as the first soft segment to provide body temperature-triggered shape memory behavior, owing to its low melting temperature[31]. Additionally, poly(ethylene glycol) (PEG) was utilized as the second soft segment to modulate hydrophilicity and mechanical properties. Considering the biocompatibility of the prepared DTPU, aliphatic isophorone diisocyanate (IPDI) was chosen for pre-polymerization. We employed the synthesized diol compound containing Diels-Alder (DA) bonds (DA-diol) as a chain extender to form a dynamic covalent network. The click-type DA reaction between furan and maleimide, known for its dynamic cross-linking mechanism without the need for catalysts or initiators[32], allowed us to build hydrogels that could be printable at high temperatures due to the thermally reversible nature of the DA reaction.

The dynamic thermoset polyurethanes (DTPUs) were synthesized in three steps (Supplementary Fig. 1). First, the chain extender DA-diol was synthesized through bismaleimide and furfuryl alcohol. Next, a series of DTPUs were prepared through two-step synthesis polymerization by adjusting the proportion of PEG and its molecular weight accordingly. The successful synthesis of DTPU was verified through Fourier-transform infrared spectroscopy (FTIR) (Supplementary Fig. 2). The absence of characteristic absorption at 2200-2280 cm$^{-1}$ demonstrated complete depletion of the excess isocyanate groups. A linear non-crosslinked polyurethane (LPU) was designed to confirm the necessity of PCL-triol for crosslinking structure. When exposed to the good solvent dimethylformamide (DMF), the LPU film, synthesized from polycaprolactone diol, completely dissolved within 30 min, whereas the DTPU only swelled (Supplementary Fig. 3), indicating the presence of a crosslinked structure. DA bonds enabled the recasting of the cut pieces of DTPU film to different shapes at 120 °C (Supplementary Fig. 4). Furthermore, the mechanical performance of the recast samples was comparable to that of the pristine samples (Supplementary Fig. 5), indicating complete recyclability. We explored the reversibility of the DA chemistry by demonstrating the retro-DA reaction of the prepared DA-diol through $^1$H NMR (Supplementary Fig. 6). Additionally, temperature-dependent FTIR was conducted to investigate the dynamic exchange capacity of DA bonds in the DTPU (Supplementary Fig. 7). The intensity of the distinctive imide peak at 690 cm$^{-1}$, which corresponds to the C=C of bismaleimide, began to increase at 100 °C, suggesting that DA bonds are relatively stable below this temperature. With the temperature increased, the DA bonds gradually dissociated. All the data from $^1$H NMR and FTIR confirmed the successful synthesis of DTPUs.

Exploiting the dynamic nature of the DA reaction, we expected the DTPU to exhibit self-healing behaviors. DTPU-0.5-4k was manually scratched using a scalpel. After 1 h of recovery at 60 °C, complete healing at the surface of the scratch was observed in optical images (Supplementary Fig. 8). Two cut specimens with different colors were reassembled to enhance visibility of cut interface. After a healing duration of 1 h at 60 °C, the healed sample sustained bending, twisting, and large strain stretching without breaking (Fig. 2a). The healing efficiencies of DTPU-0.5-4k at 25, 37, and 60 °C were 19.2%, 35.2%, and 88.9%, respectively. The sample displayed self-healing properties at

body temperature, owing to the higher chain mobility in the DTPU-0.5-4k matrix. As expected, extended healing time led to higher healing efficiency (Supplementary Fig. 9). After 12 h of healing at 37 °C, the tensile strengths of the healed samples were not significantly different from those of the pristine samples, corresponding to the high healing efficiency of 94.8%. The high-efficiency self-healing properties can facilitate interlayer adhesion of the subsequent printed structures[33].

## Mechanical properties

Tissue-matched mechanical properties of scaffolds are crucial for their applications. We comprehensively investigated the effects of the molar ratio of PEG to PCL-triol (PEG/PCL-triol) and the molecular weight of PEG ($M_{PEG}$) on the mechanical properties of DTPU before and after swelling (Fig. 2b, c). Dry samples of DTPU-0.25-1k exhibited excellent tensile strength ($\sigma_b$) of 42.5 MPa, Young's modulus ($E$) of 1.1 GPa, low breaking strain ($\varepsilon_b$) of 4.6%, and toughness ($We$) of 1.2 MJ m$^{-3}$ owing to the tight crosslinked structure (Supplementary Fig. 10). With increasing PEG/PCL-triol, the $\sigma_b$ and $E$ of DTPU-0.5-1k decreased substantially. The relatively high crosslinking density limited the activity of polymer segments, leading to brittle fracture in DTPU-1k. Conversely, an increase in $M_{PEG}$ enhanced the proportion of soft segments while reducing the cross-link density of DTPU. Consequently, DTPU-0.5-4k

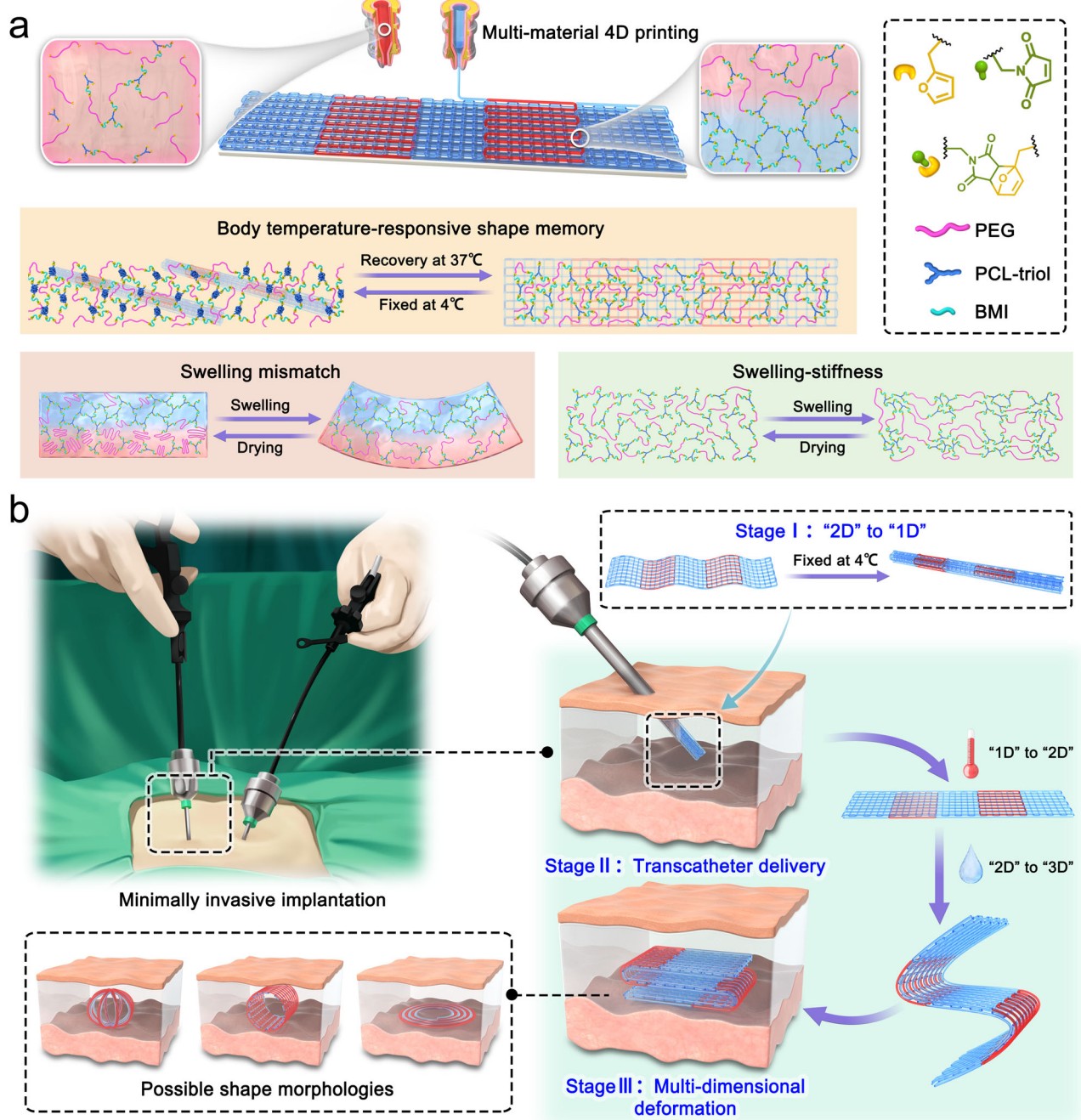

**Fig. 1 | Design and application of 4D printed scaffold. a** Multi-material 4D printing process of DTPU extruded through heated nozzles above the reversed DA reaction temperature and deposited on a substrate to form a swelling mismatch scaffold with body temperature-responsive shape memory, water-triggered programmable deformation, and swelling-stiffening properties. **b** Multi-dimensional deformation of the printed scaffold from 1D to 3D based on temperature response and water response and the operation of minimally invasive implantation.

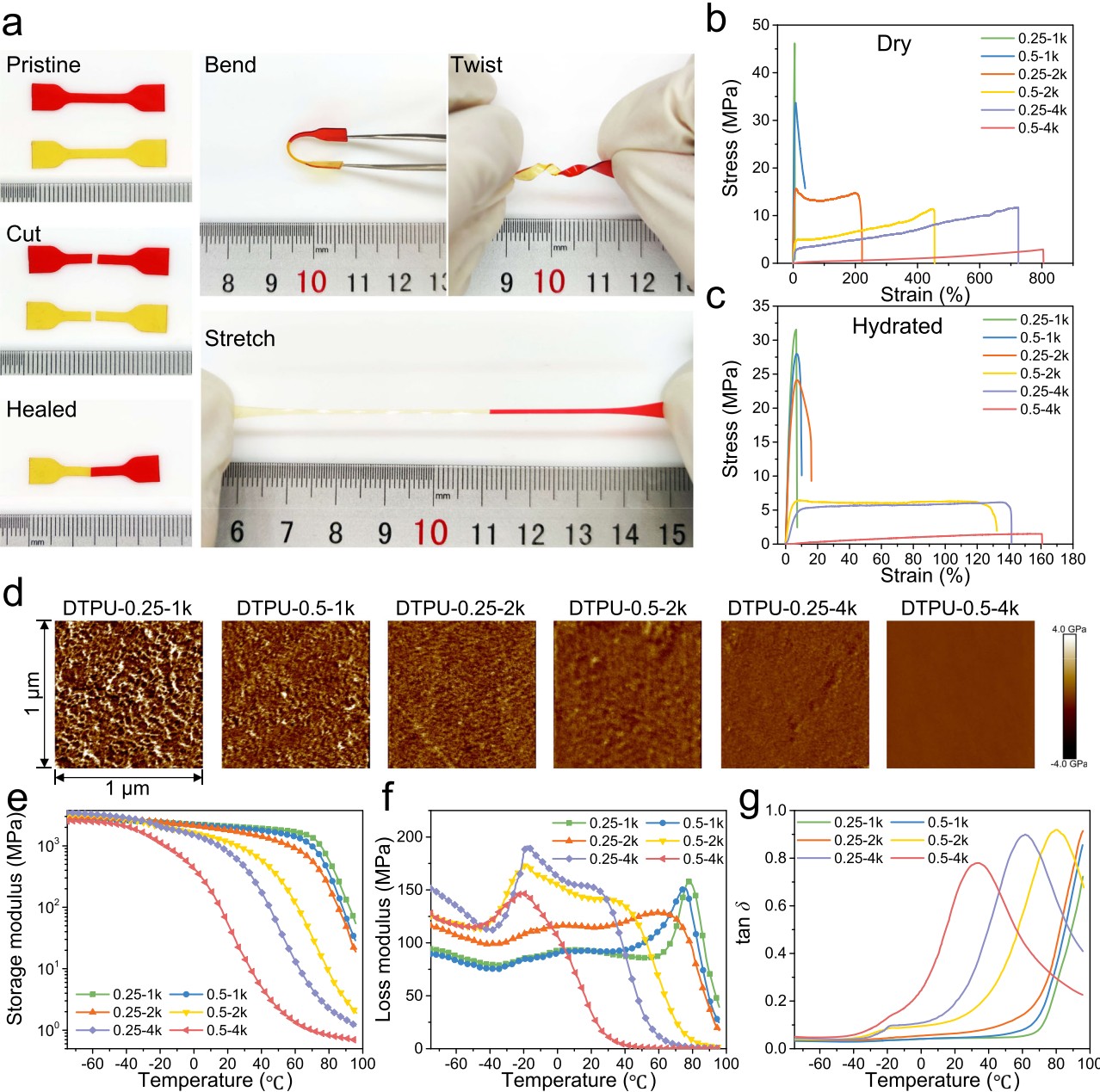

**Fig. 2 | Self-healing and mechanical performance of DTPU. a** Demonstration of self-healing between fractured surfaces through the reversible DA reaction. Two cut DTPU-0.5-4k films healed together to withstand bending, twisting, and stretching. Tensile stress strain curves of DTPU with different $M_{PEG}$ and PEG/PCL-triol in dry state (**b**), hydrated state (**c**). **d** Typical AFM images of dry DTPU films with different $M_{PEG}$ and PEG/PCL-triol ($n = 3$). Size 1 μm × 1 μm. Storage modulus-temperature curve (**e**), loss modulus-temperature curve (**f**), and tan $\delta$-temperature curve (**g**) of dry DTPU films with different $M_{PEG}$ and PEG/PCL-triol.

exhibited a large $\varepsilon_b$ of 822.3%, but relatively low $\sigma_b$ and $E$ values of 3.0 MPa and 0.7 MPa, respectively. DTPU with $M_{PEG}$ of 1k-4k and PEG/PCL-triol of 0.25-0.5 displayed remarkable mechanical properties, ranging from soft and tough to hard and brittle, indicating broad application prospects. Notably, the crosslinked structure inhibited crystallization, as evident from the absence of crystalline peaks in the X-ray diffraction (XRD) pattern (Supplementary Fig. 11), confirming the amorphous nature of DTPU.

The microstructure of DTPU films was investigated through atomic force microscopy (AFM) to further evaluate the effects of network structure on the mechanical properties of the films. Quantitative nano-mechanical mapping of DMT (Derjaguin-Muller-Toporov)

modulus revealed the influence of $M_{PEG}$ and PEG/PCL-triol on the microphase separation structures of DTPU films. The difference in modulus between the hard and soft segments caused the coexistence of prominent bright and dark areas (Fig. 2d), where the bright area reflected the clustering of hard DA segments, while the dark area corresponded to the aggregations of soft PEG and PCL segments[34,35]. DTPU-0.25-1k exhibited a distinct interface between the bright and dark areas, indicating a well-defined microphase separation structure that enhances mechanical properties. Upon increasing $M_{PEG}$ and PEG/PCL-triol, the boundaries of microphase separation became indefinite and eventually disappeared, indicating that the loose crosslinked structure weakened the aggregation of hard DA segments, and the

network tended to become more homogeneous. We speculate that the crosslinked network and microphase separation structure of DTPU films collectively contribute to their mechanical properties.

The thermal-mechanical properties were further evaluated through dynamic mechanical analysis (DMA) to confirm the changes in the microstructure of DTPU films. The initial decrease temperature of the storage modulus reversed with the increase in $M_{PEG}$ and PEG/PCL-triol, consistent with the variation trend of hard segments glass transition temperature ($T_g$) values obtained from tan $\delta$ peaks in DMA (Fig. 2e–g). The large difference in the hard segments $T_g$ values reveals that highly crosslinked network with a microphase separation structure requires higher temperatures to promote the mobility of polymer chains, implying a rigid network with high strength. Notably, the $T_g$ of the soft segments could be obtained from the loss modulus peaks in DMA, which represent a transition of the dominant motion unit in that temperature region[36,37]. The soft segments $T_g$ of DTPU-0.5 was lower than that of DTPU-0.25 because fewer constrained PEG segments were attributable to the lower crosslinking density. An increase in $M_{PEG}$ also led to a lower $T_g$ of the soft segments, as expected[38,39]. Furthermore, the hard segments $T_g$ of DTPU-0.5 was lower than DTPU-0.25 because of the decrease in hard segment content. The change in the sharpness of the loss modulus peaks indicated that the phase separation structure transitioned from hard segment-dominated to soft segment-dominated when the content of soft segments increased. Finally, the phase separation structure disappeared, consistent with the AFM results. Differential scanning calorimetry (DSC) curves also indicated that the $T_g$ of DTPU with a higher content of soft segments was more prominent than hard segment-dominated DTPU (Supplementary Fig. 12).

## Swelling-stiffening properties

The mechanical properties of hydrated DTPU were further investigated to assess water-driven phase separation. The swelling ratios of the DTPUs ranged from 11.6% to 118.1%, with increasing $M_{PEG}$ and PEG/PCL-triol, thereby promoting the hydrophilicity of the DTPUs (Supplementary Fig. 13). Ordinarily, the plasticizing effect of water would lead to a decline in $\sigma_b$, $E$, $\varepsilon_b$, and $We$ of hydrated DTPU (Supplementary Fig. 10). However, an unexpected swelling-stiffening phenomenon was observed for the DTPU with $M_{PEG} \geq 2k$. The $E$ of hydrated DTPU-0.25-2k, DTPU-0.5-2k, DTPU-0.25-4k, and DTPU-0.5-4k increased substantially, reaching 2.3, 1.3, 2.1, and 2.9 times higher than the $E$ in the dry state, respectively. Even with a swelling ratio of 118.1% for DTPU-0.5-4k, it exhibited a high $E$ of up to 2.0 MPa, indicating its classification as a stiff hydrogel[40]. The stiffness enhancement of DTPU-0.25-4k from the dry to the hydrated state was assessed through a deformation resistance test (Fig. 3a). The horizontally placed dry rectangular strip drooped under the influence of gravity when dragged by a clip. After water absorption, the hydrated strip maintained a horizontal straight state due to the stiffening induced by water. The re-dried strip bent again to similar angles, indicating the reversible nature of the swelling-stiffening property. The tensile stress-strain curve indicated that the dry and re-dried DTPU-0.25-4k possessed similar mechanical properties (Supplementary Fig. 14). Moreover, the radial resistances of two DTPU-0.25-4k stents were estimated (Fig. 3b). While the dry stents were completely flattened under a weight of 100 g, the hydrated stents withstood a 200 g weight without deformation.

Typically, the plasticizing effect of water reduces the $T_g$ value of polymers and weakens their mechanical properties. However, amphiphilic polymers facilitate water-driven phase separation, thereby enhancing mechanical performance[41]. Moreover, fewer microphase separation structures in dry DTPU were observed when $M_{PEG}$ and PEG/PCL-triol increased. Upon water absorption, hydrophilic PEG segments aggregated and separated from the hydrophobic segments because of the incompatibility of the two segments (Fig. 3c). This phenomenon is evident from the time-dependent water contact angles of the DTPU

films (Fig. 3d), which continued to decrease within the first few minutes, confirming the rearrangement and segregation of hydrophilic chains at the water contact surface[28,29]. When the $M_{PEG}$ was 1k, the movement of short PEG segments was restricted by the hard segments at both ends, resulting in the typically swelling-softening phenomenon.

The microphase separation structures of DTPU-0.25-4k in dry and hydrated states were analyzed through small-angle X-ray scattering (SAXS). The SAXS pattern of dry DTPU-0.25-4k displayed no peaks, indicating a relatively homogenous phase structure. By contrast, the pattern of hydrated DTPU-0.25-4k exhibited a broad scattering peak at $q = 0.36$ nm$^{-1}$, originating from the difference in electron cloud density between hydrophilic and hydrophobic phases, indicating an average long period of 17.45 nm for the microphase separated domains (Fig. 3e). This assumption was further confirmed by AFM results. The network structure of hydrated DTPU-0.25-4k was retained through freeze-drying. Compared with dry DTPU-0.25-4k, the lyophilized DTPU-0.25-4k showed an obvious microphase separation structure (Supplementary Fig. 15) caused by the separation of hydrophilic and hydrophobic segments. The soft segments $T_g$ of DTPU-0.25-4k decreased from −18 to −53 °C due to the plasticizing effect of water (Fig. 3f). Meanwhile, the hard segments $T_g$ increased from 62 to 70 °C, indicating that the aggregation of soft and hydrophilic PEG resulted in more densely packed hard segments. The change in $T_g$ was consistent with the SAXS and AFM results, indicating more microphase separation structure in hydrated DTPU-0.25-4k. Thus, the mobility of the chain segments was improved by lowering the crosslink density, resulting in crosslinked DTPU with swelling-stiffening properties, which are beneficial for applications in humid or aqueous environments. Although water promotes microphase separation, the hydrated DTPU with short-range order is amorphous (Supplementary Fig. 11).

## Thermal-induced shape memory and water-triggered deformation

The DTPU exhibited excellent thermal-induced shape memory performance, indicated by the sharp changes in $E$ with variations in body temperature. Figure 4a displays tensile stress-strain curves of DTPU-0.25-4k obtained at different temperatures. When the temperature changed from 10 °C to 50 °C, $\sigma_b$ and $E$ decreased from 14.6 and 70.1 MPa to 0.73 and 1.3 MPa, respectively, while $\varepsilon_b$ increased from 364.5% to 1556.3%, indicating a potential shape memory switch temperature of 37 °C. We demonstrated the temperature-dependent mechanical behavior through a DTPU-0.25-4k scaffold. At 4 °C, the scaffold could support a 100 g weight without deformation owing to its high stiffness. By contrast, at 37 °C, the support quickly flattened under the weight (Fig. 4b and Supplementary Movie 1). These results indicated that the DTPU may possess body temperature-triggered shape memory behavior.

The shape memory behaviors of the developed DTPUs in air and water were investigated. The DTPU-0.25-4k with a temporary closed-petal shape expanded into the initial 2D shape in 90 s on the heating platform at 37 °C (Supplementary Fig. 16). Moreover, the DTPU-0.25-4k with a temporary 2D shape recovered to its initial 3D shape after being heated using warm water at 37 °C, indicating shape recovery from low dimension to high dimension (Fig. 4c and Supplementary Movie 2). In the fold-deploy shape memory processes, DTPU-1k represented a relatively low shape fixation rate ($R_f$) of 90% and shape recovery rate ($R_r$) of 80% (Fig. 4d, e). However, when $M_{PEG}$ was increased, both $R_f$ and $R_r$ of the DTPU were higher than 90%, indicating excellent shape memory behavior, A reasonable explanation is that the shape memory effect of DTPU is mainly determined by the $T_g$ of hard segments, crucial for the storage modulus. As mentioned above, the highly crosslinked DTPU-1k has hard segments $T_g$ considerably higher than body temperature, with little variation in the mobility of the hard segments in the interval 4–37 °C. The increase in $M_{PEG}$ improved the mobility of the hard segments at body temperature, resulting in better shape memory behaviors.

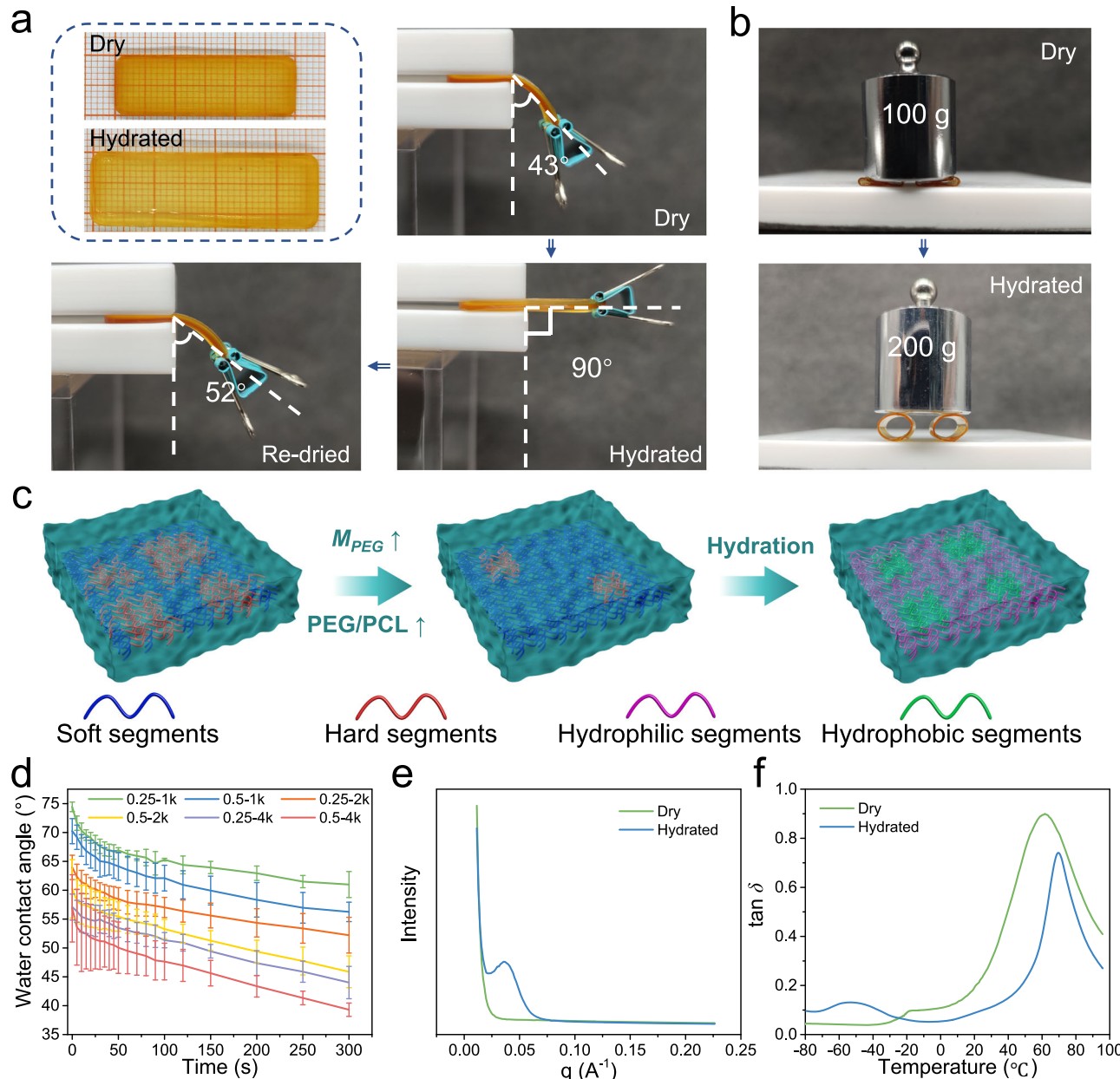

**Fig. 3 | Swelling-stiffening behavior. a** Digital photos showing the reversibility of swelling-stiffening behavior of DTPU-0.25-4k. **b** Digital photos showing swelling-induced improvement of compression resistance for DTPU-0.25-4k stents. **c** Schematic illustrations of the microphase separation structure formed between the soft and hard segments of dry DTPU and the microphase separation structure formed between the hydrophilic and hydrophobic segments of hydrated DTPU. **d** Time-dependent water contact angle of water droplet on the DTPU films with different $M_{PEG}$ and PEG/PCL-triol. **e** 1D SAXS profiles of DTPU-0.25-4k in the dry and hydrated states. **f** Tan $\delta$-temperature curves of DTPU-0.25-4k in the dry and hydrated states. Data in (**d**) are presented as mean values ± SD ($n = 3$).

Constructing swelling mismatch structures to achieve programmable deformation is a common approach to broaden the range of 4D printing polymers[12]. The swelling behaviors of the DTPUs were analyzed to guide the construction of swelling mismatch structures. The volume expansion ratios of the DTPU films from the dry to the hydrated state ranged from 22.4% to 198.5% (Fig. 4f and Supplementary Fig. 17) and positively correlated with the swelling ratios. The bilayer structure, consisting of high swelling DTPU-0.5-4k and relatively low swelling DTPU-x, exhibited controllable bending after swelling equilibrium was reached. The slender strip bent along the longitudinal axis under the internal stress resulting from swelling mismatch. The bending angles of the

bilayer strips increased from 444° to 647° with increased swelling degree differentiation (Fig. 4g, h). However, only the bilayer structures of DTPU-0.5-4k/DTPU-0.25-4k and DTPU-0.5-4k/DTPU-0.5-2k exhibited high bending angles after immersion in water for 10 min (Supplementary Fig. 18), demonstrating a rapid water response. The DTPUs undergo rapid water-triggered deformation upon achieving swelling equilibrium quickly (Supplementary Fig. 13). We constructed a multi-level structure composed of three different bilayers. After being immersed in water for 10 min, the structure deformed to a flower shape via a pre-programmed path (Fig. 4i), confirming the feasibility of deformation programming based on swelling mismatch.

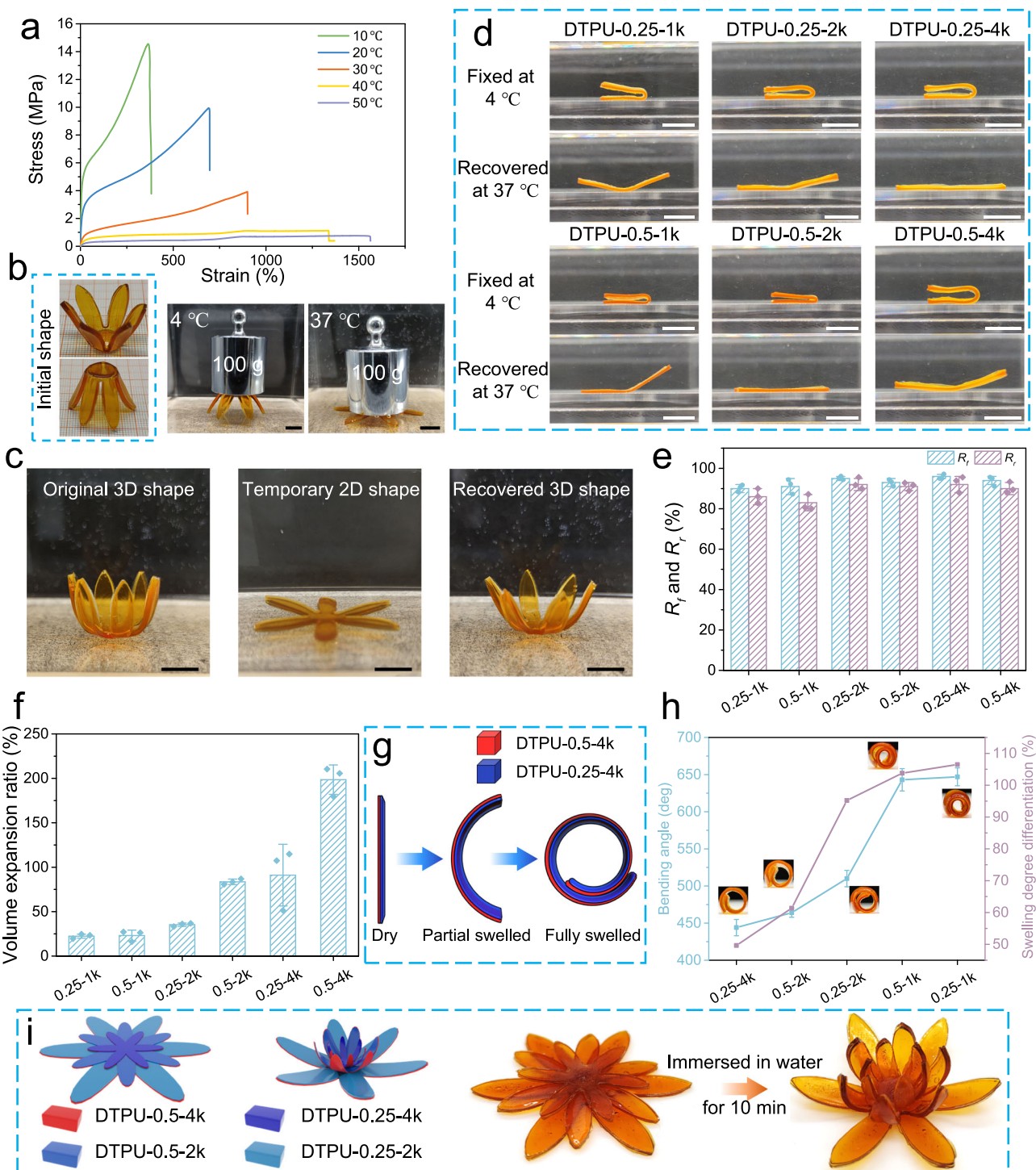

**Fig. 4 | Thermal-induced shape memory and water-triggered bending deformation of DTPU. a** Tensile stress strain curves of DTPU-0.25-4k at various temperatures in the 10 to 50 °C range. **b** Digital photos showing the capacity of DTPU-0.25-4k to resist deformation at 4 °C and 37 °C, respectively. **c** Visual demonstration of the complete thermal-induced shape memory and recovery processes of DTPU-0.25-4k petal. **d** The fold-deploy shape memory behavior of DTPU films. **e** $R_f$ and $R_r$ of the DTPU films corresponding to the fold-deploy shape memory method.

**f** Volume expansion ratio of DTPU in hydrated state compared to the corresponding dry DTPU. **g** Schematics of bending deformation mechanism of the bilayer constructed by two DTPU films with different swelling degrees. **h** The final bending angles of swelling mismatch structures with different swelling degree differentiation. **i** Diagram of the designed three-layered 2D flat petal structure and the predicted water-triggered deformation after immersed in water for 10 min. Data in **e**, **f**, and **h** are presented as mean values ± SD. ($n$ = 3). Scale bars:10 mm.

## Printability

To evaluated the printability of the DTPUs through widely-used extrusion-based FDM, the rheological behaviors were tested and optimized to ensure that DTPUs can be extruded through a nozzle and deposited at the designed position to form complex architectures.

The thermal reversibility of the DA reaction enabled all the DTPUs to transition from an elastic state (shear storage modulus ($G'$) > shear loss modulus ($G''$)) to a viscous liquid state ($G'' > G'$) when the temperature increased (Fig. 5a), endowing the covalently crosslinked networks with thermoplastic properties. The crossover between $G'$ and $G''$ occurred

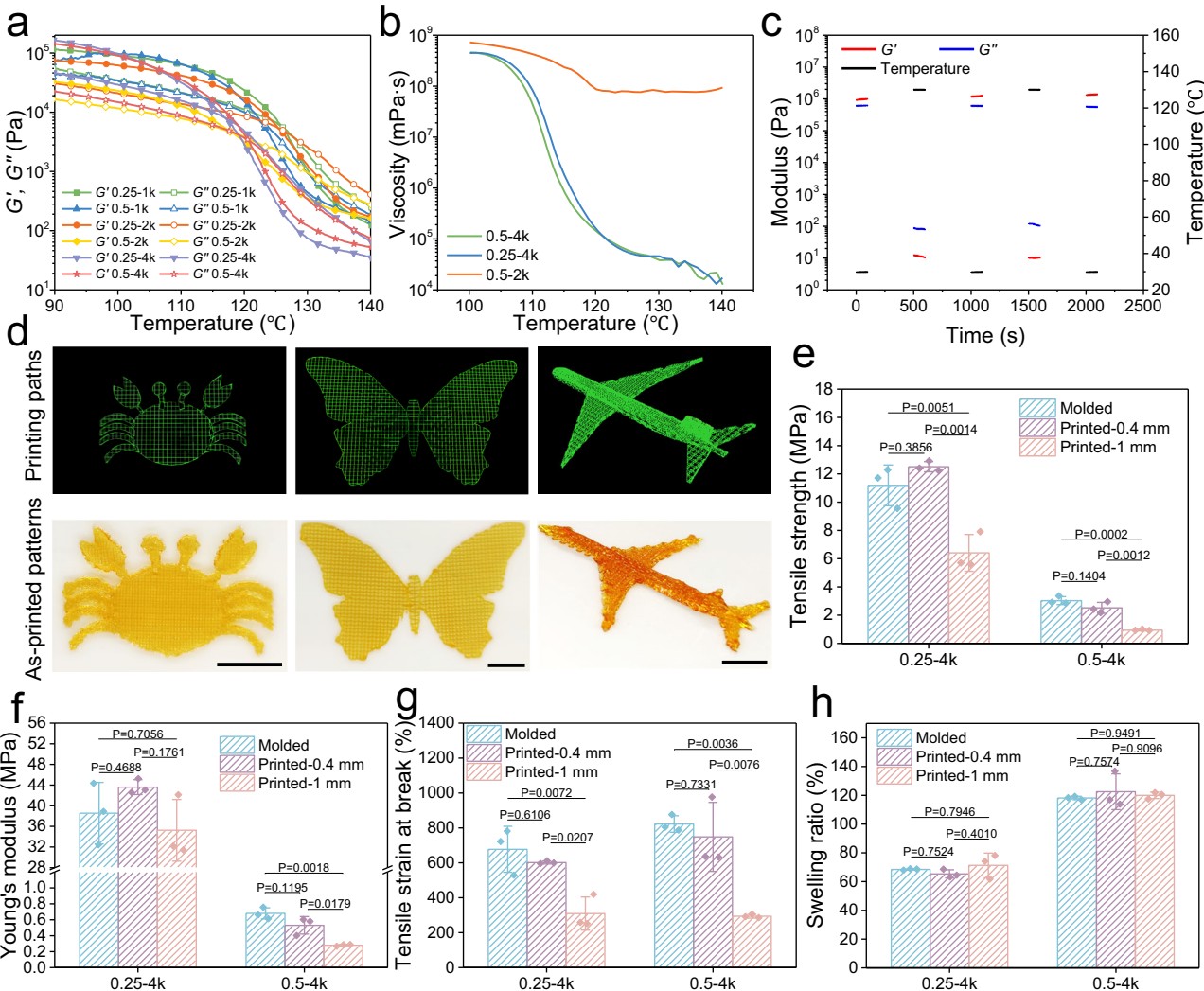

**Fig. 5 | Printability of DTPUs. a** Oscillatory temperature sweep of dry DTPUs with different $M_{PEG}$ and PEG/PCL-triol showing transition temperature between elastic state and viscoelastic liquid state. **b** Viscosity of DTPU-0.5-2k, DTPU-0.25-4k, and DTPU-0.5-4k as a function of temperature. **c** Oscillatory alternating temperature sweep of DTPU-0.25-4k. **d** Printing paths and as-prepared structures printed using DTPU-0.25-4k. Tensile strength **e**, Young's modulus **f**, and tensile strain at break **g**, of dry scaffolds constructed with different molding methods. **h** Swelling ratio of scaffolds constructed with different molding methods. Values in (**e**–**h**) represent mean ± SD. One-way analysis of variance (ANOVA), Tukey's post hoc test. ($n = 3$).

below 130 °C, indicating the possibility of extrusion. Thermogravimetric analysis (TGA) revealed that the DTPU did not decompose below 240 °C (Supplementary Fig. 19), indicating that 130 °C is a safe printing temperature. Moreover, viscosity is another important parameter that affects extrudability because high viscosity coupled with high pneumatic extrusion pressure, which can result in difficulties during printing[42]. Owing to their high water response speed, DTPU-0.5-2k, DTPU-0.25-4k, and DTPU-0.5-4k were selected to evaluate the changes in viscosity (Fig. 5b). At 130 °C, DTPU-0.5-2k exhibited a relatively high viscosity of $8 \times 10^4$ Pa s, while DTPU-0.25-4k and DTPU-0.5-4k exhibited low viscosities of 46 and 38 Pa s, respectively, which are suitable for extrusion. By contrast, print ink with high viscosity achieves good shape fidelity by avoiding the formation of tension-driven droplets during extrusion[43]. Compared with DTPU-0.25-4k and DTPU-0.5-4k, the grid structure printed with DTPU-0.5-2k had a line diameter of 351 μm (Supplementary Fig. 20), which was closest to the nozzle diameter of 300 μm, indicating optimal shape fidelity. Considering similar viscosity and shape fidelity, DTPU-0.25-4k and DTPU-0.5-4k were selected for multi-material 4D printing. To further test the temperature thixotropy, DTPU-0.25-4k and DTPU-0.5-4k were subjected to alternating low (30 °C) and high (130 °C) temperatures. Rapid

and reversible transitions from elastic state to viscous liquid state were observed, indicating repeatable printing capability of DTPU (Fig. 5c and Supplementary Fig. 21).

Intricate structures such as a butterfly, crab, and airplane were customized using FDM to evaluate the 3D printing performance of DTPU-0.25-4k (Fig. 5d). To analyze the effect of the print path on mechanical performance, three types of 2-layer stripes printed with different filament orientations were prepared. The topography of the printed structures clearly reflected the initial configuration (Supplementary Fig. 22a). Remarkably, the parallel and orthogonal samples exhibited similar $\sigma_b$ of 11.9 and 12.2 MPa, $E$ of 47.8 and 46.1 MPa, and $\varepsilon_b$ of 597.1% and 597.4%, respectively, indicating robust adhesion between sequentially deposited layers (Supplementary Fig. 22b). The two printed structures exhibited higher stiffness than the molded sample, likely because of the orientation of microfilaments[44]. The excellent ductility indicated that the printed structure had few defects. However, the perpendicular sample only exhibited $\sigma_b$ of 4.4 MPa, $E$ of 38.2 MPa, and $\varepsilon_b$ of 187.2%, respectively, indicating stacking defects between intralayer microfilaments. Thus, DTPU printed through an orthogonal path can effectively maintain the mechanical performance of extrusion-printed structures.

To comprehensively examine the effect of printing procedures on the above-mentioned thermal-induced shape memory, swelling-stiffening behavior, and water-triggered deformation, DTPU-0.25-4k and DTPU-0.5-4k were used to print scaffolds through an orthogonal path with different line distance. The dry scaffolds with 0.4 mm line distance showed mechanical properties comparable to those of the cast molded samples (Fig. 5e–g). However, the mechanical properties of the dry scaffolds printed with 1 mm line distance were impaired. For example, dry DTPU-0.5-4k scaffold printed with1 mm line distance showed $\sigma_b$ of 0.9 MPa, $E$ of 0.3 MPa, and $\varepsilon_b$ of 294.0%, respectively, significantly lower than those of the dry cast DTPU-0.5-4k samples ($\sigma_b$ of 3.0 MPa, $E$ of 0.7 MPa, and $\varepsilon_b$ of 822.3%). The mechanical deterioration is attributed to the pore structure within the printed scaffold, reducing the volume of material dissipating energy. In contrast, no pore existed in the scaffold with 0.4 mm line distance due to the thick line diameter (Supplementary Fig. 20). The $\varepsilon_b$ of the hydrated scaffolds decreased significantly, indicating that the print defects were amplified during the swelling of printed samples (Supplementary Fig. 23). While the maintenance of $\sigma_b$ and $E$ was attributed to the absence of pore in samples printed at 0.4 mm line distance. We assumed that the retention of $E$ for hydrated scaffolds printed with 1 mm line distance is due to swelling resulting in a smaller pore structure, thereby increasing the material volume for dissipating energy in the stretching direction. In short, the printing parameter of line distance has an impact on the mechanical properties of scaffolds due to the presence of the pore structure. However, the $E$ of hydrated scaffolds were significantly higher than those of the corresponding dry scaffolds, indicating that the printing procedure does not affect the swelling-stiffening behavior.

The storage modulus of the printed scaffolds decreased with the increase in printing line distance, which is consistent with the influence of line distance on mechanics (Supplementary Fig. 24a). However, the peak of tan $\delta$ in the low-temperature region disappeared (Supplementary Fig. 24b), which is because the rapid temperature drop after extrusion printing results in insufficient time to form an obvious phase separation structure. After water absorption, the hydrated scaffolds exhibited tan $\delta$ peaks at −50 °C (Supplementary Fig. 24c), indicating that swelling led to microphase separation formed between hydrophilic and hydrophobic chain segments. Based on the sharp change in the storage modulus of the scaffolds near body temperature, the printed butterfly and crab were fixed into a vivid temporary shape at 4 °C. In a 37 °C environment, the 3D temporary shape quickly reverted to the original flat pattern (Supplementary Fig. 25), demonstrating that the body temperature-triggered shape memory function of the printed structure is not affected. Finally, the influence of the printing process on the swelling ratio was determined. Benefiting from the pore structures[45], the printed scaffolds displayed shorter balancing time within 5 min (Supplementary Fig. 26). While the final swelling ratios were basically consistent with those of cast molded samples (Fig. 5h), indicating that the printing process has no effect on the swelling ratio of DTPU. Therefore, it is feasible to construct swelling mismatch structures by multi-material printing.

**Multi-material 4D printing**

Stress mismatch between the extruded microfilaments with different compositions, based on different volume expansion ratios, enables the programmable deformation of multi-material 4D printed structures. Two-layer grids with different print paths were printed to demonstrate programmable deformation (Fig. 6a). The passive layer was printed using DTPU-0.25-4k in a parallel or perpendicular direction. Then, the active layer was printed using DTPU-0.5-4k at an orientation angle $\theta$ relative to the long axis of the passive layer. After swelling in water, the enlarged grids bent along the microfilaments in the active layer, with the passive layer facing inside. The final 3D structures transformed from printed flat patterns were determined by the oblique angle $\theta$ of

the microfilaments in the active layer. At $\theta = 0°$ or 90°, the printed patterns deformed into tubes with different lengths and bending angles, indicating that the deformation amplitude is affected by the length of the microfilaments in the active layer[46]. At $\theta = 45°$, the printed patterns morphed into right-handed cylinder helices, consistent with the expected deformation results.

To further evaluate the feasibility of constructing complex 3D scaffolds, different flat patterns were fabricated through multi-material 4D printing (Fig. 6b). Similarly, the bottom layer was printed using DTPU-0.25-4k. For the top layer, DTPU-0.5-4k was used to print the pre-programmed deformation part, while the passive part was printed using DTPU-0.25-4k. After the rapid swelling process, the printed flat patterns transformed into pre-determined 3D structures, such as self-standing 3D domes and curved barriers. Notably, same-layer multi-material 4D printing can be realized using FDM owing to its accuracy and the excellent printability of DTPUs. A two-layer strip was designed with both areas printed using DTPU-0.25-4k and DTPU-0.5-4k at the bottom layer. The top layer was constructed orthogonally by printing with another ink corresponding to the bottom layer. While the active layers were printed in different areas of the top and bottom layers, the strip curled in opposite directions at each end after swelling in water, resulting in a double roll. The capacity of the 4D printed structures to deform from low to high dimensions represents the possibility of minimally invasive treatment.

The feasibility of using DTPU as minimally invasive implantable scaffolds was investigated through in vitro experiments conducted for proof of concept. Figure 6c depicts a 2D helical structure mimicking the fibrous ring of the intervertebral disc. The printed 1D bilayer line with a 0.9 mm diameter bent radially upon exposure to water and deformed into a 2D helical shape with an external diameter of 15 mm. Figure 6d displays a long strip with a diameter of 4 mm, formed by bonding the ends of printed bilayer double cross pattern. Upon hydration, the eight edges bent outward from the center to form a 3D hollow spherical scaffold with a diameter of 18 mm, facilitating the minimally invasive filling of soft tissue defects. Figure 6e illustrates the minimally invasive implantation of a printed 2D sheet as a vascular stent. The sheet was curled and fixed into a thin cylindrical shape, with the active layer facing inside. After delivery to the vascular site, the cylinder expanded in response to the stimulation of warm blood and supported the vessel. Incorporating a water-triggered response may exert a stronger chronic outward force on the stent compared with conventional temperature-responsive deployment, preventing restenosis of the target lesion. Figure 6f illustrates a more complex 3D folded structure wherein each layer does not necessarily bend, while the junction between adjacent layers must bend in opposite directions to achieve folding. The 3D scaffold with a folded structure can be applied to the minimally invasive filling of cartilage defects. It is worth mentioning that all printed scaffolds can complete body temperature-triggered shape recovery and water-responsive programmable deformation within 10 min.

Further, the printed scaffolds were fixed in different ways to examine the effect of simultaneous temperature and water responses on the final shape (Supplementary Fig. 27). Whether inwards (DTPU-0.25-4k layer facing inside) or outwards (DTPU-0.5-4k layer facing inside), fixing by folding or curling the 2D pattern around the short axis did not affect the final pre-determined 3D shape. However, slight misalignment was found in the final 3D shape when fixing by curling around the long axis. It is highly plausible due to the excessively high water-response rate, as this type of fixing causes an uneven rate of water absorption between inside and outside, which leads to tilting when bending. Attributed to the porous nature of the printed structure, the slight misalignment showed little impact on the final 3D shape.

The integration of multi-endogenous stimuli responsiveness and minimally invasive implantation was demonstrated through

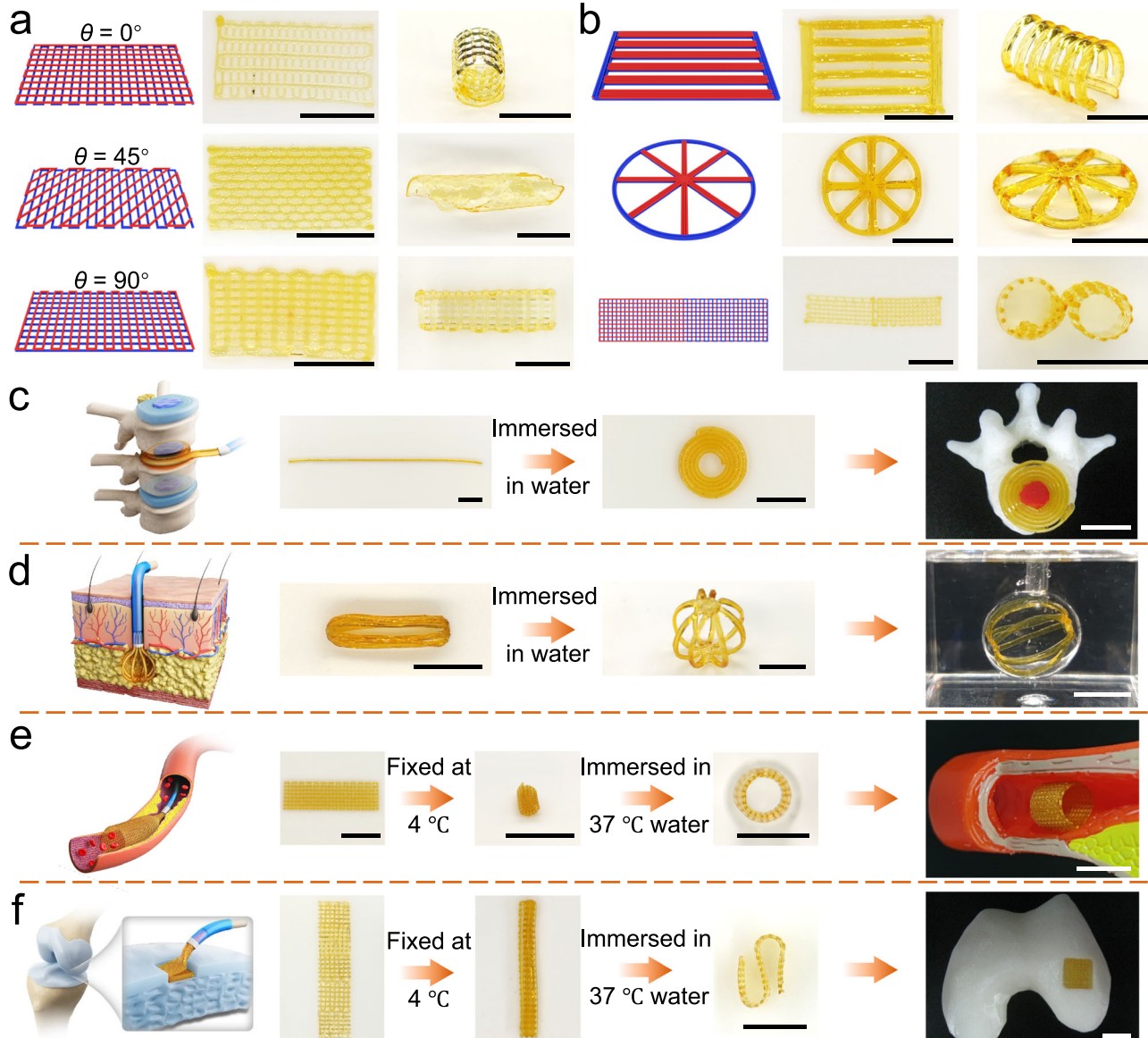

**Fig. 6 | Multi-material 4D printing and potential biomedical applications.**
**a** Programmable deformation of the multi-material 4D printed two-layer grids with different alignments of the active layer. **b** Programmable deformation of designed 2D flat patterns. The left column, middle column, and right column in (**a**, **b**) represented the initial configuration, the as-printed 2D patterns, and the deformed 3D structures, respectively. **c**–**f** Schematic illustrations and photographs of printed structures undergoing multi-dimensional deformation and potential applications as fibrous ring replacement (**c**), soft tissue defect support (**d**), vascular scaffolds (**e**), and cartilage defect scaffolds (**f**). The deformation in (**c**), and (**d**) involves programmable deformation in response to water. The deformation in (**e**), and (**f**) involves body temperature-triggered shape memory and programmable deformation in response to water. Scale bars:10 mm.

in vitro and in vivo experiments (Fig. 7a). In the in vitro test, a printed 2D sheet was rolled into a 1D rod around the long axis and placed in water for 10 s at 4 °C. Then, the 1D rod was manually placed on the proximal end of a catheter with a 3.5 mm inner diameter and delivered through a guidewire. After delivery to 37 °C water, the 1D rod unfolded quickly and curled around the short axis, exhibiting pre-programmed deformation (Fig. 7b and Supplementary Movie 3). Next, the feasibility of the 4D printed scaffold to complete the transformation from 1D to 3D shapes was verified through in vivo endogenous stimulation (Fig. 7c and Supplementary Movie 4). Water at 37 °C was injected under the skin of Sprague Dawley (SD) rats to create deformation space and allow the scaffold to swell homogeneously, and the temporary 1D rod was implanted subcutaneously through a catheter. After stimulation for 3 min, the skin was cut to retrieve the deformed 3D scaffold. The retrieved scaffold exhibited a structure remarkably close to the desired one.

Hence, the scaffold achieved pre-programmable deformation in vivo once sufficient space was available.

Moreover, cytotoxicity tests confirmed that all the DTPU films were non-toxic to L929 cells (Supplementary Fig. 28). And the scaffolds printed with DTPU-0.25-4k or DTPU-0.5-4k also demonstrated excellent biocompatibility. To investigate the foreign body response, scaffolds printed with DTPU-0.25-4k or DTPU-0.5-4k were subcutaneously implanted in SD rats. On week 2 and week 4, the scaffolds were completely covered by the surrounding tissue, and no obvious difference around the tissue could be seen (Fig. 7d). Histological evaluation through hematoxylin-eosin staining also showed a postoperative inflammatory response with accumulations of inflammatory cells in sham-control and experimental groups after 1 week (Fig. 7e). But the inflammatory response gradually disappeared at 2 and 4 weeks, indicating good in vivo biocompatibility of the DTPU scaffolds. Therefore, the developed DTPUs provided a strategy for minimally invasive

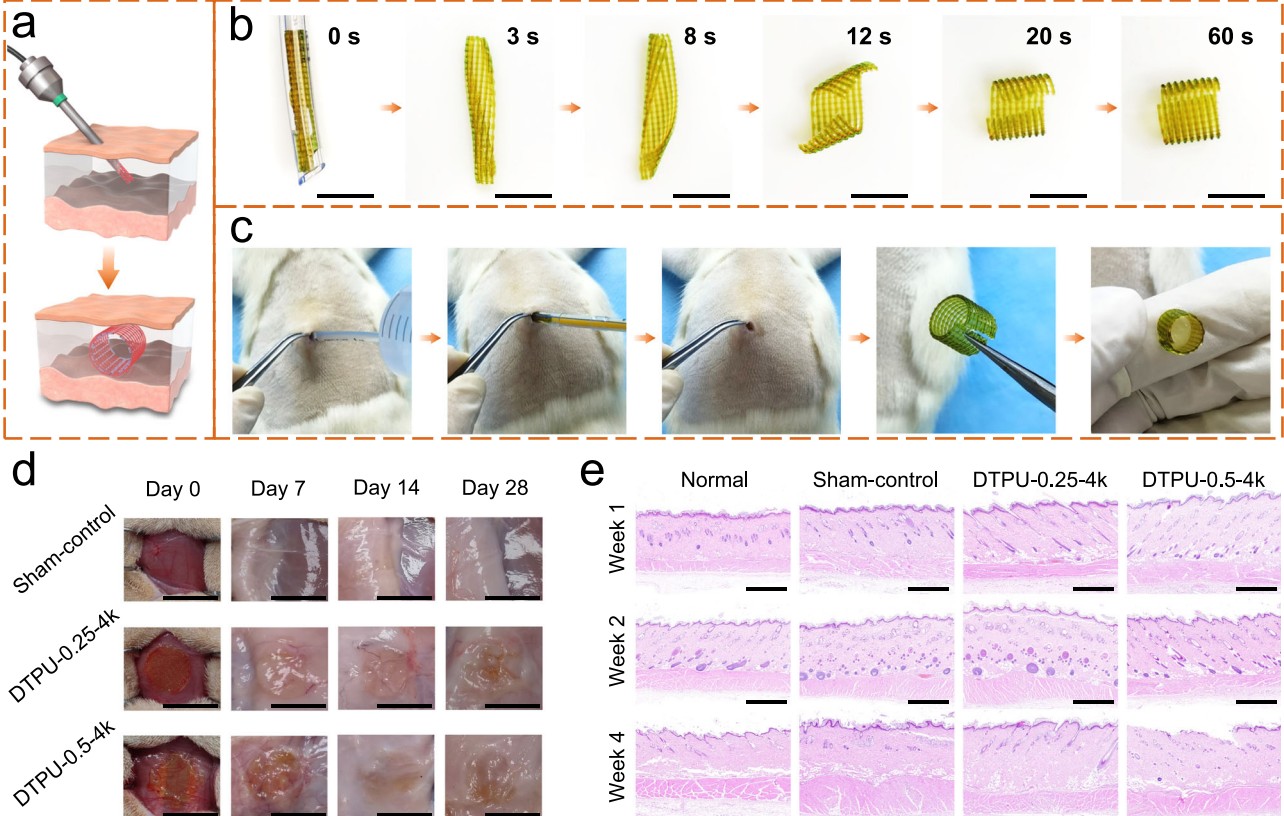

**Fig. 7 | Transcatheter delivery and biocompatibility. a** Schematic of 4D printed scaffold for transcatheter delivery and subcutaneous multi-dimensional deformation. **b** Digital images revealing the transcatheter delivery procedure and multi-endogenous stimuli responsiveness of the 4D printed structure. **c** Photographs showing the feasibility of 4D printed scaffolds to complete 1D to 3D deformation subcutaneously in SD rats. **d** Gross pictures of the implanted scaffolds and surrounding tissues at different time points. **e** Histological evaluation of the printed DTPU-0.25-4k and DTPU-0.5-4k scaffolds at different timepoints (*n* = 3). Scale bars in (**b**) and (**d**) were 10 mm, and scale bars in (**e**) were 1 mm.

delivery of void-filling scaffolds, showing potential in biomedical applications.

## Discussion

We developed a mechanically tunable and dynamically crosslinked shape memory polyurethane system, which is compatible with extrusion-based multi-material 4D printing to fabricate scaffolds with body temperature-triggered shape memory function, water-triggered programmable deformation, and swelling-stiffening properties. The DTPU system mainly consists of PCL-triol and PEG as soft segments and DA-diol as a chain extender. The thermally reversible nature of the DA reaction endows printability to the DTPU. Furthermore, we performed comprehensive experiments to investigate the effect of polymer composition on mechanical properties, temperature-sensitive shape memory behavior, water-triggered programmable deformability, and swelling-stiffening properties. Meanwhile, in vitro and in vivo experiments revealed that the printed structure could be fixed into a 1D shape for catheter delivery. Once implanted in the body, it transformed spontaneously into the desired 3D shape for cavity filling and mechanical support under the stimulations of body temperature and body fluids while circumventing harmful stimuli. The multi-dimensional deformability and swelling–stiffening properties of 4D printed scaffolds make them suitable for minimally invasive treatment of tissue defects.

## Methods
### Materials
All chemicals used in this study were purchased from Macklin with AR grade or higher (unless otherwise specified) and used as received. Dibutyltin dilaurate (DBTDL, 95%) was purchased from Sigma-Aldrich. Tetrahydrofuran (THF, 99.9%, superdry) and N,N-dimethylformamide (DMF, 99.9%, superdry) were purchased from J&K Scientific Ltd. Teflon molds of various shapes and depths were manufactured in-house to shape polyurethane.

### Diels-alder diol (DA-diol) preparation
N,N'-4,4'-diphenylmethane-bismaleimide (0.05 mol) was dissolved in DMF at 60 °C in a 100 mL three-necked glass flask. Furfuryl alcohol (0.15 mol) was added to the flask, followed by stirring and refluxing for 3 h to facilitate the reaction. Subsequently, the solvent was added dropwise into cold diethyl ether to precipitate the products. The yellow products were washed three times with diethyl ether and dried under vacuum at 60 °C.

### Diels-Alder dynamic crosslinked polyurethane preparation
PCL-triol (1.1 g, 2 mmol) and an appropriate amount of PEG were heated at 120 °C in a 100 mL three-necked glass flask for 1 h under vacuum while stirring mechanically to remove water. Then, a mixture of THF (20 mL) and DMF (4 mL) was added to the flask at 60 °C. Under a nitrogen atmosphere, IPDI and DBTDL (20 μL) were added, and the pre-polymerization was allowed to occur by stirring and refluxing for 3 h. Moreover, the as-prepared chain extender DA-diol was dissolved in DMF, and the mixture was added to the prepolymer solution. After a chain extension reaction for another 5 h, the solution was transferred into a Teflon mold and placed in a fume hood overnight. After the solvents evaporated, the polyurethane films were vacuum-dried in a 60 °C oven for 1 day, synthesizing a series of Diels-Alder dynamic crosslinked polyurethane films. To eliminate the consumption of the

isocyanate groups by traces of water in the monomer and solvent, the molar ratio of the isocyanate group to hydroxyl (-NCO/-OH) was 1.06. The excess -NCO inside the DTPU were completely depleted during the post-processing. The polyurethanes were denoted as DTPU-x-y, where x represents the PEG/PCL-triol, and y indicates the molecular weight of PEG. To recast the polyurethane film, the existing film was cut into pieces and mixed with DMF. The mixture was heated at 120 °C for 20 min to ensure the complete dissolution of DTPU. Then, the solution was transferred into the Teflon mold and placed in a 60 °C oven for 2 days to remove DMF. To prepare linear non-crosslinked polyurethane (LPU), PCL-triol was replaced with PCL-diol, while the other reaction conditions were identical.

## Characterization

The chemical composition of DTPU was analyzed through an attenuated total reflection Fourier transform infrared spectrometer (Invenio R, Bruker, Germany) in the range of 4000-400 cm$^{-1}$. The temperature-dependent absorbance FTIR spectra of the DTPU-0.5-2k film were recorded at a 4 cm$^{-1}$ spectral resolution using a Bruker Tensor 27 spectrometer over a temperature range from 30 to 140 °C at a heating rate of 3 °C min$^{-1}$ to detect the dynamic reversibility of Diels-Alder bonds. $^1$H NMR spectra were obtained using a Bruker AVANCE III 400 spectrometer with dimethyl sulfoxide (DMSO-d$_6$) as the solvent to assess the dynamic reversible DA chemistry of DA-diol. The spectrum of retro-DA was recorded immediately after DA-diol was heated at 120 °C for 20 min.

## Mechanical testing

The tensile strength of the DTPU was measured using a mechanical test system (IBTC-300S, Tianjin) at room temperature at a stretch speed of 50 mm min$^{-1}$. The variable temperature tensile test was performed using an electromechanical dynamometer (Legend 2344, Instron, USA) with a water bath to control the temperature. Before testing, DTPU films were cut to a standard size using a IV dumbbell cutter that conformed to the GB/T 528-2009 standard. The printed samples were cut into regular rectangles with dimensions approximately 20 mm × 5 mm × 0.1 mm. The results of three separate tests were averaged to assess the mechanical properties. $E$ was determined from the slope of the stress-strain curve with the initial linear portion below 10% strain. The toughness of the DTPUs was calculated by integrating the area under the stress-strain curves. For the scratch-healing test, the prepared DTPU film (30 mm × 10 mm × 0.5 mm) was attached to a slide by double-sided tape. And a scratch was created at the center of the film by using a scalpel. Subsequently, the slide carrying the film was placed in 60 °C oven for varying periods of time to examine the self-healing. To quantify the healing efficiency, the prepared DTPU films were cut into dumbbell-shaped specimens using a IV dumbbell cutter that conformed to the GB/T 528-2009 standard. And the dumbbell-shaped specimens were cut in half with a blade. The two parts were immediately reattached and placed under different conditions for healing. No external force was applied to the specimens during the entire healing process. The healing efficiency was determined using Eq. (1):

$$\text{Healing efficiency} = \frac{\sigma_{b1}}{\sigma_{b0}} \times 100\% \tag{1}$$

where $\sigma_{b1}$ denotes the tensile strength of the healed sample and $\sigma_{b0}$ denotes the tensile strength of the pristine sample.

## Dynamic mechanical analysis (DMA)

The storage modulus, loss modulus, and loss factor of DTPU were measured using a DMA-Q800 (TA, USA) in the tension (film) mode. The dynamic temperature sweep was performed at 0.1% strain, 1 Hz frequency, a preload of 0.01 N, and a heating rate of 5 °C min$^{-1}$ from −80 °C to 100 °C. The dimensions of the cast molded specimens were

approximately 12 mm × 4 mm × 0.5 mm. The dimensions of the printed specimens were approximately 15 mm × 4 mm × 0.1 mm.

## Rheological measurement

Rheological behaviors of DTPU were evaluated using a rheometer (MCR-302, Anton-Paar, Austria) equipped with a parallel steel plate of a 25-mm diameter to confirm the printability of DTPU. DTPU films were cut into a disk shape with a diameter of 25 mm and thickness of 1 mm before testing. Oscillatory temperature sweeps were implemented from 90 °C to 140 °C (heating rate: 5 °C min$^{-1}$) with 1% strain amplitude and 1 Hz frequency. Oscillatory alternating temperature sweeps were performed between 30 °C and 130 °C. The total time for each temperature change and retention was set as 400 s. And each measurement time was fixed as 100 s with 1 Hz. Viscosity measurements were carried out with temperature ramped from 100 °C to 140 °C at a shear rate of 1 s$^{-1}$ and a heating rate of 5 °C min$^{-1}$.

## Thermogravimetric analysis (TGA)

The thermal stability of DTPU was determined using a thermal analyzer instrument (TG 209 F3, Netzsch, Germany) in the temperature range from 40 °C to 800 °C with a heating rate of 20 °C min$^{-1}$ under a N$_2$ atmosphere. All the samples were lyophilized and dehydrated before testing.

## Differential scanning calorimetry (DSC)

DSC experiments were performed on DSC-Q2000 (TA, USA) in the temperature range from −80 °C to 100 °C at a programmed ramp of 10 °C/min under N$_2$ atmosphere. Before the experiments, DTPU was heated to 60 °C and kept for 5 min to eliminate the heat history.

## Atomic force microscope (AFM)

Quantitative nano-mechanical measurements were performed using a Dimension Icon AFM (Bruker, Germany) in PeakForce QNM mode to record elastic modulus maps. The measurements were performed under ambient conditions. Silicon-based RTESPA-300 was used as the AFM probe. The deflection sensitivity of the probe was calibrated using a standard sapphire sample with a known Young's modulus, and the spring constant of the cantilever was 40 N/m. The curvature radius of the tip was approximately 10 nm. The scanning frequency of 0.5 Hz was used within a scan area, and the scan size was 1 × 1 μm$^2$ for all measurements. To prevent operation on damaged sample surfaces or repeated measurements of identically contacted points, a scan resolution of 256 × 256 points was selected, ensuring that the minimum distance between neighboring depressions was appreciably larger than the diameter of the probe contact.

## X-ray diffraction (XRD)

The crystal degrees of the DTPUs were analyzed using an X-ray diffractometer (D8 Advance, Bruker, Germany) with Cu Kα radiation (λ = 0.154 nm) in the 2θ range from 10° to 50° at a scanning speed of 4° min$^{-1}$. The DTPUs adhered to the sample stage and were either dry or hydrated (equilibrium swelling) before testing.

## Small angle X-ray scattering (SAXS)

SAXS measurements were conducted using a Cu Kα X-ray radiation source (NanoSTAR, Bruker, Germany) with a radiation wavelength of 0.154 nm. The operating voltage and current were 50 kV and 0.6 mA, respectively. The DTPU-0.25-4k films (10 mm × 5 mm) with different thicknesses (0.4 mm for dry film, 0.7 mm for hydrated film) were fixed on the clear aperture of the sample stage and then exposed to X-ray for 10 min to capture images.

## Time-dependent water contact angle

The time-dependent water contact angles were measured with an optical tensiometer (Theta Lite, Biolin Scientific, Finland). 4 μL distilled

water droplet was dripped on the DTPU films at 25 °C through a pipettor. And the images taken at different times were recorded. The results of three separate tests were averaged to assess the time-dependent water contact angles.

### Swelling ratio

The swelling test of the DTPUs was performed at 37 °C in deionized water. Briefly, dry DTPU films were weighed and put into deionized water. Each film was taken out at regular time intervals and weighed after removing the excess water from the film surface. The swelling ratio was determined using Eq. (2):

$$\text{Swelling ratio} = \frac{m_t - m_0}{m_0} \times 100\% \qquad (2)$$

where $m_0$ denotes the original mass of the film (dry) and $m_t$ denotes the mass of the film after swelling in water at time t.

### Volume expansion ratio

To calculate the volume expansion ratio of the DTPUs, dry DTPU films with the same shape were placed in deionized water at 37 °C for 2 days, and the deionized water was refreshed every 12 h. The volume of the film after swelling equilibrium was assessed. The volume expansion ratio was determined using Eq. (3):

$$\text{Volume expansion ratio} = \frac{V_1 - V_0}{V_0} \times 100\% \qquad (3)$$

where $V_0$ denotes the original volume of the film (dry) and $V_1$ denotes the volume of the hydrated film after swelling equilibrium.

### Thermal-induced shape memory behavior

The shape memory ability of DTPU was quantitatively examined through the fold-deploy shape memory method. The stripe-shaped samples (30 mm × 10 mm × 1 mm) were folded into the same angle in a water bath at 37 °C for 10 s and then fixed in an ice water bath for 5 s. The angles between the two blades were measured and recorded as $\theta_f$. Finally, the specimens with temporary shapes were immersed in water at 37 °C for 5 min, after which the angles between the blades were measured again and recorded as $\theta_r$. The shape fixity ratio ($R_f$) and the shape recovery ratio ($R_r$) were calculated using Eqs. (4) and (5), respectively.

$$R_f = \frac{180° - \theta_f}{180°} \times 100\% \qquad (4)$$

$$R_r = \frac{\theta_r}{180°} \times 100\% \qquad (5)$$

The initial 2D petal film was first fixed into a 3D closed bud to examine the ability of DTPU to change from a 3D shape to a 2D shape under temperature stimuli. Subsequently, the temporary 3D bud was placed on a 37 °C heating table to observe the transition from a 3D shape to a 2D shape.

The DTPU with 3D shapes (bud or hollow ball) was first pressed and fixed into a flat 2D film by applying an external force to examine the ability of DTPU to change from a 2D shape to a 3D shape under temperature stimuli. Then, the temporary 2D film was placed in a 37 °C water bath to observe the transition from a 2D shape to a 3D shape.

### Swelling mismatch-induced bending deformation

To investigate the effect of swelling mismatch on the degree of bending deformation, the stripe-shaped DTPU-0.5-4k (30 mm × 10 mm × 0.5 mm) was overlapped on different DTPU strips and placed in a 90 °C oven for 4 h to form a tightly bonded bilayer structure. Then, the bilayer was immersed in 37 °C deionized water, and the bending behavior was recorded at different times. The final bending angle was calculated according to the central angle obtained by drawing two lines between the endpoints of the curly strip and the core of the circle.

A three-layered 2D flat petal structure was constructed to demonstrate the effects of swelling mismatch on the degree of deformation. Each layer of the petal structure comprised two layers of DTPU with different swelling ratios. The small bilayer petal was composed of 0.5-mm thick DTPU-0.5-4k and DTPU-0.25-4k with a diameter of 27 mm. The medium bilayer petal comprised 0.5-mm thick DTPU-0.5-4k and DTPU-0.5-2k with a diameter of 36 mm. The large bilayer petal was composed of 0.5-mm thick DTPU-0.25-4k and DTPU-0.5-2k with a diameter of 45 mm. The three-layered structure was immersed in 37 °C deionized water for 10 min, and then the shape deformation was recorded.

### Multi-material 4D printing

The printing was performed using an organ printing united system (Novaprint, China). The pneumatic extrusion process used micro-nozzles with an inner diameter of 300 μm. For all the DTPUs, the printing temperature was set at 130 °C, while the base plate was set at 25 °C. Other printing parameters included a printing speed of 500 mm/min, a printing air pressure of 200 kPa, and a line distance of 1 mm, unless specifically stated. All 3D models were designed using 3ds Max software.

### Cytotoxicity assay

The cast DTPU films and the printed 2D scaffolds were equilibrated in PBS and sterilized using ultraviolet for 0.5 h. Sterilized DTPU hydrogels were immersed in Roswell Park Memorial Institute medium 1640 (RPMI-1640) for 72 h at 37 °C to prepare the leachates. The volume of RPMI-1640 equaled 1 mL per 0.1 g of the equilibrated DTPU. Mouse fibroblast (L929) cells were seeded in 96-well plates at a density of $2 \times 10^4$ cells/well and incubated for 24 h at 37 °C in a 5% $CO_2$ humidified atmosphere. Then, the culture medium was replaced with 100 μL complete growth medium and 100 μL leachate. After 24 h incubation, the medium was removed and 100 μL medium containing MTT (0.5 mg/mL) was added into the wells and the incubation was continued for 4 h. The medium was removed and 150 μL dimethyl sulfoxide was added to dissolve formed formazan crystals. After shaking for 10 min, the spectrophotometric readings were operated at 490 nm by a Multiskan GO plate reader (Thermo Fisher Scientific, USA). Cells cultured in extracts without material samples, extracts of HD-poly-ethylene, and phenol solution, were considered as blank, negative, and positive controls, respectively. Cell viability was calculated as a relative percentage of living cells compared to the blank control.

### In vivo transcatheter delivery test

Three male SD rats (6-week-old, 200 ± 10 g) were purchased from Zhuhai BesTest Bio-Tech Co., Ltd. (China). All the protocols for animal experiments were performed according to the Guide for the Care and Use of Laboratory Animals 8th edition (National Institutes of Health, Nat. Acad. Press, 2011) with project approval from the Institutional Animal Care and Use Committee of the Shenzhen Institutes of Advanced Technology, Chinese Academy of Sciences (approval numbers: SIAT-IACUC-190328-YYS-ZXL-A0727). Before the test, anesthesia was induced in the SD rat with isoflurane using a gas anesthesia machine (RWD, R640). Then, a 2 mm diameter incision was cut on the back of the depilated rat. The 4D printed scaffold was fixed into a temporary 1D shape and delivered subcutaneously to the rat via a 3.5 mm diameter catheter. An appropriate amount of 37 °C water was injected under the skin simultaneously. After 3 min, the water was removed with a syringe. The skin was cut along the incision to retrieve the scaffold. Finally, the rat was euthanized by $CO_2$ asphyxiation.

## In vivo biocompatibility test

To examine the in vivo biocompatibility of DTPU, the printed disc scaffolds with 8 mm diameter were implanted under the skin of SD rats. All the animal experiments were complied with the guidelines of Tianjin Medical Experimental Animal Care, and animal protocols were approved by the Institutional Animal Care and Use Committee of Yi Shengyuan Gene Technology (Tianjin) Co., Ltd. (protocol number: YSY-DWLL-2023252). The printed scaffolds were sterilized using ultraviolet for 2 h and equilibrated in sterile PBS. Subsequently, nine male SD rats (6-week-old, 200 ± 10 g) purchased from Beijing HFK BIOSCIENCE Co., Ltd. (China) were anesthetized with isoflurane using a gas anesthesia machine (ZS-MV-IV). The scaffolds printed with DTPU-0.25-4k or DTPU-0.5-4k were implanted under the skin. At 1-week, 2-week, and 4-week timepoints, the scaffolds along with the surrounding tissues were excised and fixed with 4% paraformaldehyde for 48 h. The local interaction of the implanted scaffolds with the tissues were assessed by histological evaluation based on hematoxylin-eosin (H&E) staining. The sham-control group was set to eliminate the inflammatory response induced by the open wound.

## Statistics and reproducibility

Statistical analysis was performed with GraphPad Prism9.5. Statistical significance between the two groups was calculated using a two-tailed $t$-test. Multiple comparisons were assessed using one-way analysis of variance (ANOVA) with Tukey post-test. Three independent experiments were carried out and data were expressed as mean ± standard deviation.

## Reporting summary

Further information on research design is available in the Nature Portfolio Reporting Summary linked to this article.

## Data availability

The data generated in this study are provided in the Supplementary Information/Source Data file. The full image dataset is available from the corresponding author on request. Source data are provided with this paper.

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

## Acknowledgements

This work was supported by the National Key Research and Development Program of China (2018YFA0703100, Grant recipient: X.Z.), the National Natural Science Foundation of China (52203151, 32201125, 81972071, Grant recipient: B.L., S.B., X.Z.), China Postdoctoral Science Foundation (2021M703361, Grant recipient: B.L.), and Guangdong Basic and Applied Basic Research Foundation (2021A1515110794, 2021A1515110902, Grant recipient: B.L., S.B.), and the Science and Technology Research Funding of Shenzhen (JCYJ20200109150420892, JCYJ20210324102014039, Grant recipient: S.B.). We thank to Dr. Yang Wang from the Bruker (Beijing) Scientific Technology Co. Ltd. for the AFM observation and analysis of the data.

## Author contributions

X.Z., H.P., and B.L. designed the project. B.L., W.L., W.W.L., J.W., and S.B. contributed to design experiments and discuss results. B.L., H.L., F.M., Z.X., L.H., H.Z., and C.W. conducted the experiments. B.L., X.Z., and W.L. wrote the manuscript. B.L., F.M., Y.Y., and X.Z. participated in the revision of the manuscript. All authors edited and agreed with the manuscript.

## Competing interests

The authors declare no competing interests.
