## [Peer Review File · Nature Communications]

4D printed hydrogel scaffold with swelling-stiffening properties and programmable deformation for minimally invasive implantationREVIEWER COMMENTS

Reviewer #1 (Remarks to the Author):

In this manuscript, the authors prepared polyurethane-based hydrogel ink (PUHI), and the inks can be printed into scaffolds with body temperature-triggered shape memory and water-triggered programmable deformation using multi-material 4D printing. The authors introduce a thermally reversible dynamic covalent bond to impart viscoelastic rheological behavior for extrusion printing and enhance interlayer adhesion between the constituent layers. This work does not present novelty in terms of either a remarkable finding or deep insights warranting publication in a top journal. Scientifically, several statements are incorrect/inappropriate, the experimental section is unexplained and not clearly presented, and additional experiments are required to support some of the claims. Specific comments are as follows.

1. Printing was performed at 130°C using extrusion-based printing. Did the authors use the synthesized PU solution? If yes, then what is the percentage, and what is the solvent? Since the PU was printed directly in its melt state, so how can this be a hydrogel scaffold?
2. The author mentioned in the paper title that “4D printed hydrogel scaffold”, but it is unclear how this polymer qualifies as a “hydrogel”. The photographs of the printed material do not resemble a hydrogel.
3. PUHI was synthesized in DMF. Is the synthesized prepolymer solution soluble in water? If not, then how could it be the hydrogel ink? If yes, then what is the water percentage to make PU hydrogel ink, and how would it be possible to print that ink at 130°C?
4. Supplementary Fig 9 C, the elongation and fracture strength of the pristine tensile stress-strain curve is less than the healing curve of 12 h, which means that healing efficiency is more than 100%. Can the author explain why elongation and fracture strength are more for the healed sample than the original?
5. Were all the studies for DMA, Mechanical, swelling, shape memory performed for PUHI i.e, hydrogel ink? Does the printed scaffold also show the same property? Since the final application is based on the printed scaffold, authors should study the above-mentioned properties of printed scaffolds, not the hydrogel ink. There is no value in studying the ink.
6. What is the molar ratio between NCO and di-ol? As per the reaction scheme, NCO functional group will remain in the PUHI system, then how do the authors avoid the toxicity of the isocyanate group?
7. In figure 2 f, why the loss modulus increased near -20°C and decreased at high temperatures is unclear. Can the author explain these phenomena?
8. Since the authors are using extrusion-based printing, in Figure 2, the plot of viscosity with temperature and the thixotropy test at various temperatures would be more appropriate than storage and loss modulus plots.
9. What happens if a deformed or fixed structure is exposed to temperature and water simultaneously, and what would the recovery and final shape programmability be? Then, will recovery and programmability of the final shape be affected? Author should study this.

Reviewer #2 (Remarks to the Author):

This is an excellent manuscript dealing with memory shapes polymers how 4d printing properties. The manuscript is very well written. The mechanical properties are well supported by AFM, chemical analysis and structural properties. The visual material is very useful.

My comments are the following

- 1) No statistics for the measurements are provided in the case mechanical characterization and healing efficiency. Can the authors comment on that and demonstrate that the measurements are reproducible?
- 2) The text under the SFM images and the scale bar are barely visible. Also, the quality of the images probably needs to be improved.
- 3) For self healing please provide more details on how the cut was performed.

Overall, I believe this is an excellent manuscript which can be published after minor revisions

Reviewer #3 (Remarks to the Author):

This paper characterized the development of novel shape memory polymers for use with 3D printing applications, for the generation of implants that would require minimally invasive surgeries. The work presented is novel, and is of interest to the scientific community. The authors did a wonderful job fully characterizing their materials - as far as mechanical, swelling, and thermal-induced shape memory. I recommend this paper for publication following minor revisions:

Since the paper is built around potential in vivo applications I think the paper fell a little short on cytotoxicity assays, the authors did evaluate toxicity generated by leachables following a 2 day incubation of their material in media - I feel a better evaluation would be to follow: ISO 10993-12:2021(E) (Biological evaluation of medical devices — Part 12: Sample preparation and reference materials). While in vivo data may be beyond the scope of this communication, that would also have been satisfying to see what kind of foreign body response the material induces.

Minor edits:

Figure 2 a - replace "cutted" with cut,

Figure 3a - please include an image of the printed part before manipulation

Figure 3c - could you present the actual AFM data? What is the scale?

Figure 4b include an image of the original part without weight manipulation

Question for the authors: for the in vivo experiment - why was the 37C water flush necessary? Could the

researchers just waited longer or have placed a heating pad over the animal?
Do each of the shapes have a specific time scale for when they fully transform at 37C?

Point-by-point response to the reviewers' comments:

Response to Reviewer #1:

Comments: In this manuscript, the authors prepared polyurethane-based hydrogel ink (PUHI), and the inks can be printed into scaffolds with body temperature-triggered shape memory and water-triggered programmable deformation using multi-material 4D printing. The authors introduce a thermally reversible dynamic covalent bond to impart viscoelastic rheological behavior for extrusion printing and enhance interlayer adhesion between the constituent layers. This work does not present novelty in terms of either a remarkable finding or deep insights warranting publication in a top journal. Scientifically, several statements are incorrect/inappropriate, the experimental section is unexplained and not clearly presented, and additional experiments are required to support some of the claims. Specific comments are as follows.

Response: We appreciate your constructive suggestions to our manuscript, and we have carefully revised the manuscript to ensure that we clearly describe the significance of our work and strive to highlight the unique aspects of our study. For your suggestion of additional experiments to support certain claims, we have carefully considered and incorporated any necessary experiments to strengthen the scientific rigor of our work.

In short, we constructed a mechanically supported hydrogel scaffold with good biocompatibility, excellent mechanical strength, and responsiveness to two mild endogenous stimuli, which enables minimally invasive implantation and can effectively reduce the trauma size.

The novelty could be highlighted as following points:

1. *Realization of multi-dimensional deformation (1D to 3D) of printed scaffolds in response to mild endogenous stimuli (body temperature and body fluid) for addressing the concern of large trauma caused by the implantation process of*

conventional printed scaffolds.

Since 4D printed scaffolds can change shape and function in response to external stimuli, they are expected to be used in minimally invasive tissue-engineered applications [1-3]. There are articles utilizing 4D printing technology to achieve minimally invasive implantation of tissue-engineered scaffolds. And the shape transformation of the scaffolds in most articles is one-dimensional (from 2D to 3D) [3-7]. However, 2D shaped scaffolds are difficult to implant directly through narrow minimally invasive surgical channels [8], showing limited ability to reduce trauma size. A few studies have constructed scaffolds implanted in temporary 1D structures based on shape memory function [9]. But the 3D shape recovery requires external stimuli such as high temperature and magnetic field, which may impair tissue and organ function [10].

In this work, we used the prepared dynamic thermoset polyurethane to construct scaffolds with both body temperature and body fluid response via multi-material 4D printing technology. Through integration of temperature-sensitive shape memory and swelling mismatch, the printed 2D flat patterns can be fixed into a temporary 1D roll-up shape, facilitating transcatheter delivery. After immersion in 37 °C water, the 1D roll recovers to the initial 2D pattern and absorbs water to achieve programmed deformation, finally forming a 3D hydrogel scaffold. This procedure demonstrates the realization of minimally invasive implantation of 3D scaffolds with complex printed structures based on endogenous stimulation, which may significantly reduce the surgical trauma.

2. Biocompatible high-strength hydrogel scaffolds for mechanical support.

Hydrogels are considered to be the most promising scaffold material for tissue engineering due to their excellent swelling properties and similarity to soft tissues [11,12]. Printed scaffolds based on hydrogels have been widely used in biomedical applications [13-15]. However, hydrogel inks that can be used for extrusion printing typically exhibit weak mechanical properties. And the printed hydrogel scaffolds show

worse mechanical properties [2], which is unfavorable for transcatheter delivery of the hydrogel scaffolds, as the scaffolds may be fragmented during delivery [16]. While printed hydrogel scaffolds with high strength usually require harmful stimuli to achieve shape memory [17-19].

In this work, we used the biocompatible poly(ethylene glycol) (PEG) and poly(ϵ -caprolactone) triol (PCL-triol) as soft segments to construct amphiphilic dynamic thermoset polyurethanes (DTPUs), which can form a phase-separated structure upon water absorption and lead to mechanical enhancement. The swelling-stiffening property facilitates the biomedical application of hydrogel scaffolds in high humidity environments, where flexible scaffolds are desired for minimally invasive delivery while maintaining rigidity for mechanical support after implantation. Although the printed scaffolds are water-free during implantation, they eventually are deployed in the body as hydrogel scaffolds after water-triggered programmable deformation.

We summarized a table to compare the reported study about 4D printed scaffolds for minimally invasive implantation with our study.

Table 1. Comparison with other 4D printed scaffolds in literature

Materials	Endogenous stimulus	Number of stimuli-responsive	Stimuli	Transition	Hydrogel	Reference
Alg and MC; ADA-Gel; Sodium acrylate; HEA and EPOX	Yes	1	Water	2D to 3D	Yes	5, 20-22
PU dispersion, Gel, and GelMA	Yes	1	37 °C	2D to 3D	Yes	23
PGDA-sugar; Bisphenol A diglycidyl ether, poly(propylene glycol) bis(2-amino propyl) ether, and decylamine; Polycarbonate	Yes	1	35-37 °C	2D to 3D	No	3, 24, 25
AuNPs/nHA/SMPU; MM3520; TCP/P(DLLA-TMC)	No	1	40-45 °C	2D to 3D	No	4, 26-30
Agarose/AM/Laponite	No	1	95 °C	2D to 3D	Yes	31
AA-MA and HA-MA	No	1	Ca ²⁺	2D to 3D	Yes	32
CS and CA	No	1	Ethanol	2D to 3D	Yes	33
PLA; PLA/Fe ₃ O ₄ ; E-51-PUP and Fe ₃ O ₄ @GO	No	2	>50 °C Magnetic	1D to 3D	No	9, 34-38
MA-BSA, NIPAAm, and DMAEMA	No	3	pH 2-10 50 °C Enzymatic	2D to 3D	Yes	39
DTPU	Yes	2	37 °C/Water	1D to 3D	Yes	This work

- [1] Kuang, X. et al. Advances in 4D printing: Materials and applications. *Adv. Funct. Mater.* **29**, 1805290 (2019).
- [2] Champeau, M. et al. 4D printing of hydrogels: A review. *Adv. Funct. Mater.* **30**, 1910606 (2020).
- [3] Weems, A. C., Arno, M. C., Yu, W., Huckstepp, R. T. R. & Dove, A. P. 4D polycarbonates via stereolithography as scaffolds for soft tissue repair. *Nat. Commun.* **12**, 3771 (2021).
- [4] Deng, Y. et al. 4D printed shape memory polyurethane-based composite for bionic cartilage scaffolds. *ACS Appl. Polym. Mater.* **5**, 1283-1292 (2023).
- [5] Joshi, A. et al. 4D printed programmable shape-morphing hydrogels as intraoperative self-folding nerve conduits for sutureless neurorrhaphy. *Adv. Healthc. Mater.* **12**, 2300701 (2023).
- [6] Lai, J. et al. 4D printing of highly printable and shape morphing hydrogels composed of alginate and methylcellulose. *Mater. Design* **205**, 109699 (2021).
- [7] Luo, Y., Lin, X., Chen, B. & Wei, X. Cell-laden four-dimensional bioprinting using near-infrared-triggered shape-morphing alginate/polydopamine bioinks. *Biofabrication* **11**, 045019 (2019).
- [8] Li, S. et al. In vivo self-assembled shape-memory polyurethane for minimally invasive delivery and therapy. *Mater. Horiz.* **10**, 3438-3449 (2023).
- [9] Lin, C. et al. 4D-printed biodegradable and remotely controllable shape memory occlusion devices. *Adv. Funct. Mater.* **29**, 1906569 (2019).
- [10] Miao, S. et al. 4D printing of polymeric materials for tissue and organ regeneration. *Mater. Today* **20**, 577-591 (2017).
- [11] Naahidi, S. et al. Biocompatibility of hydrogel-based scaffolds for tissue engineering applications. *Biotechnol. Adv.* **35**, 530-544 (2017).
- [12] Rosales, A. M. & Anseth, K. S. The design of reversible hydrogels to capture extracellular matrix dynamics. *Nat. Rev. Mater.* **1**, 15012 (2016).
- [13] Choi, S. et al. Fibre-infused gel scaffolds guide cardiomyocyte alignment in 3D-printed ventricles. *Nat. Mater.* **22**, 1039-1046 (2023).
- [14] Lee, A. et al. 3D bioprinting of collagen to rebuild components of the human heart. *Science* **365**, 482-487 (2019).
- [15] Yu, C. et al. Photopolymerizable biomaterials and light-based 3D printing strategies for biomedical applications. *Chem. Rev.* **120**, 10695-10743 (2020).
- [16] Zhang, Y. et al. Radiopaque highly stiff and tough shape memory hydrogel microcoils for permanent embolization of arteries. *Adv. Funct. Mater.* **28**, 1705962 (2018).
- [17] Wang, Z., Heck, M., Yang, W., Wilhelm, M. & Levkin, P. A. Tough PEGgels by in situ phase separation for 4D printing. *Adv. Funct. Mater.* 2300947 (2023).
- [18] Zhou, Q., Yang, K., He, J., Yang, H. & Zhang, X. A novel 3D-printable hydrogel with high mechanical strength and shape memory properties. *J. Mater. Chem. C* **7**, 14913-14922 (2019).
- [19] He, Y. et al. Digital light processing 4D printing of transparent, strong, highly conductive hydrogels. *ACS Appl. Mater. Interfaces* **13**, 36286-36294 (2021).
- [20] Kitana, W., Apsite, I., Hazur, J., Boccaccini, A. R. & Ionov, L. 4D biofabrication

- of T-shaped vascular bifurcation. *Adv. Mater. Technol.* **8**, 2200429 (2023).
- [21] Hiendlmeier, L. et al. 4D-printed soft and stretchable self-folding cuff electrodes for small-nerve interfacing. *Adv. Mater.* **35**, 2210206 (2023).
- [22] Schwartz, J. J. & Boydston, A. J. Multimaterial actinic spatial control 3D and 4D printing. *Nat. Commun.* **10**, 791 (2019).
- [23] Wu, S. D. & Hsu, S. 4D bioprintable self-healing hydrogel with shape memory and cryopreserving properties. *Biofabrication* **13**, 045029 (2021).
- [24] Bond, G. et al. 4D printing of biocompatible, hierarchically porous shape memory polymeric structures. *Biomater. Adv.* **153**, 213575 (2023).
- [25] Hann, S. Y., Cui, H., Esworthy, T. & Zhang, L. G. 4D thermo-responsive smart hiPSC-CM cardiac construct for myocardial cell therapy. *Int. J. Nanomedicine* **18**, 1809-1821 (2023).
- [26] Deng, Y. et al. 4D printed orbital stent for the treatment of enophthalmic invagination. *Biomaterials* **291**, 121886 (2022).
- [27] Deng, Y. et al. Programmable 4D printing of photoactive shape memory composite structures. *ACS Appl. Mater. Interfaces* **14**, 42568-42577 (2022).
- [28] Kashyap, D., Kumar, P. K. & Kanagaraj, S. 4D printed porous radiopaque shape memory polyurethane for endovascular embolization. *Addit. Manuf.* **24**, 687-695 (2018).
- [29] Hendrikson, W. J. et al. Towards 4D printed scaffolds for tissue engineering: exploiting 3D shape memory polymers to deliver time-controlled stimulus on cultured cells. *Biofabrication* **9**, 031001 (2017).
- [30] Wang, C. et al. Advanced reconfigurable scaffolds fabricated by 4D printing for treating critical-size bone defects of irregular shapes. *Biofabrication* **12**, 045025 (2020).
- [31] Guo, J., Zhang, R. & Zhang, L. 4D printing of robust hydrogels consisted of agarose nanofibers and polyacrylamide. *ACS Macro Lett.* **7**, 442-446 (2018).
- [32] Kirillova, A., Maxson, R., Stoychev, G., Gomillion, C. T. & Ionov, L. 4D biofabrication using shape-morphing hydrogels. *Adv. Mater.* **29**, 1703443 (2017).
- [33] Parimita, S., Kumar, A., Krishnaswamy, H. & Ghosh, P. Solvent triggered shape morphism of 4D printed hydrogels. *J. Manuf. Process.* **85**, 875-884 (2023).
- [34] Zhao, W., Li, N., Liu, L., Leng, J. & Liu, Y. Origami derived self-assembly stents fabricated via 4D printing. *Compos. Struct.* **293**, 115669 (2022).
- [35] Chapuis, J. N. & Shea, K. Redeployable, 4D printed wave spring actuators. *Mater. Design* **232**, 112163 (2023).
- [36] Peng, W. et al. 4D printed shape memory anastomosis ring with controllable shape transformation and degradation. *Adv. Funct. Mater.* **33**, 2214505 (2023).
- [37] Wei, H. et al. Direct-write fabrication of 4D active shape-changing structures based on a shape memory polymer and its nanocomposite. *ACS Appl. Mater. Interfaces* **9**, 876-883 (2017).
- [38] Ma, B. et al. 4D printing of multi-stimuli responsive rigid smart composite materials with self-healing ability. *Chem. Eng. J.* **466**, 143420 (2023).
- [39] Narupai, B., Smith, P. T. & Nelson, A. 4D printing of multi-stimuli responsive protein-based hydrogels for autonomous shape transformations. *Adv. Funct. Mater.* **31**, 2011012 (2021).

Revision: The introduction was revised to describe the novelty more clearly. *In Page 3 Line 65 (Introduction):* “Nevertheless, printed hydrogel scaffolds with mechanical support and multidimensional morphing (1D to 3D) in response to mild endogenous stimuli have not been reported yet.” *In Page 3 Line 70 (Introduction):* “thus broadening the range of print materials for constructing hydrogel scaffolds [16, 24, 25]. The integration of shape memory polymer and multi-material 4D printing offers viable ideas for multidimensional morphing under the mild internal environment.” *In Page 3 Line 76 (Introduction):* “An alternative hydrogel with soft-stiff transition is needed to balance the deformation during transplantation and mechanical support at the targeted site. Recently, some amphiphilic polymers achieving water-induced stiffening based on water-driven phase separation have been reported [28-30], providing a new idea for constructing swelling-stiffening hydrogel scaffolds.” *In Page 3 Line 81 (Introduction):* “Herein, we report an approach to deliver scaffolds in a minimally invasive manner through multidimensional morphing (1D to 3D) by the development of amphiphilic dynamic thermoset polyurethane (DTPU). Thermally reversible dynamic covalent bonds were introduced to impart viscoelastic rheological behavior for extrusion printing and enhance interlayer adhesion between the constituent layers. Through multi-material 4D printing based on fused deposition modeling (FDM), the DTPU can be printed into 2D scaffolds with body temperature-triggered shape memory and water-triggered programmable deformation (Fig. 1a).”

Q1. Printing was performed at 130°C using extrusion-based printing. Did the authors use the synthesized PU solution? If yes, then what is the percentage, and what is the solvent? Since the PU was printed directly in its melt state, so how can this be a hydrogel scaffold?

Response: We thank the reviewer for this suggestion. The prepared dry dynamic thermoset polyurethane (DTPU) without any solution was sheared and put into a cylinder for fused deposition printing. The printed DTPU scaffolds are water-free. Because of the swelling mismatch design, the printed DTPU scaffolds were expected

to absorb water *in vivo* and deform to the desired 3D structure to provide support, in which case the swollen three-dimensional cross-linking network became a hydrogel. Since the high resemblance between hydrogels and extracellular matrix [1,2], we had introduced hydrophilic PEG chain segments to the DTPU and expected the prepared scaffolds to function in a hydrogel state *in vivo* after swelling. We apologize for the incorrect description of “hydrogel inks”, etc. in the manuscript. We have changed the “hydrogel ink” to “dynamic thermoset polyurethane”. And only the scaffolds that accomplished water-responsive deformation were referred to as “hydrogel scaffolds” in the revised manuscript.

[1] Naahidi, S. et al. Biocompatibility of hydrogel-based scaffolds for tissue engineering applications. *Biotechnol. Adv.* **35**, 530-544 (2017).

[2] Rosales, A. M. & Anseth, K. S. The design of reversible hydrogels to capture extracellular matrix dynamics. *Nat. Rev. Mater.* **1**, 15012 (2016).

Revision: We have changed the “polyurethane-based hydrogel ink” and “polyurethane-based hydrogel inks” to “dynamic thermoset polyurethane” and “dynamic thermoset polyurethanes”, respectively. All the “PUHI” and “PUHIs” have been changed to “DTPU” and “DTPUs”, respectively. And only the scaffolds that accomplished water-responsive deformation were referred to as “hydrogel scaffolds” in the revised manuscript. *In Page 3 Line 90 (Introduction):* “After immersion in 37 °C water, the 1D roll recovers to the initial 2D pattern and absorbs water to achieve programmed deformation, finally forming a 3D hydrogel scaffold.”

Q2. The author mentioned in the paper title that “4D printed hydrogel scaffold”, but it is unclear how this polymer qualifies as a “hydrogel”. The photographs of the printed material do not resemble a hydrogel.

Response: We apologize for the unclear description of the “4D printed hydrogel scaffold”. Generally, a hydrogel is defined as a water-containing polymer network which can be cross-linked by chemical or physical bonds [1,2]. In this work, the dynamic thermoset polyurethane (DTPU) scaffolds obtained after high temperature

extrusion printing are chemically cross-linked polymers that do not contain water. We have demonstrated the chemically cross-linked structure of the DTPU in Supplementary Figure 3, which could only swell but not dissolve in good solvent dimethylformamide (DMF). While the linear thermoplastic polyurethane prepared with PCL-diol could fully dissolve in DMF within 30 min. Furthermore, the DTPU exhibits the ability to absorb water due to the introduction of PEG (Supplementary Fig. 13). For example, the DTPU-0.25-4k and DTPU-0.5-4k used for multi-material 4D printing exhibit the swelling ratio of 68.5% and 118.1%, respectively. Some studies have used the prepared amphiphilic polyurethane hydrogels for biomedical applications [3-5]. Since the prepared scaffold is in hydrogel state after deployment *in vivo* wet environment, we refer to it as a 4D printed hydrogel scaffold in the title.

[1] Yuk, H., Lu, B. & Zhao X. Hydrogel bioelectronics. *Chem. Soc. Rev.* **48**, 1642-1667 (2019).

[2] Yang, C. & Suo, Z. Hydrogel ionotronics. *Nat. Rev. Mater.* **3**, 125-142 (2018).

[3] Oveissi, F., Naficy, S., Le, T. Y. L., Fletcher, D. F. & Dehghani, F. Tough hydrophilic polyurethane-based hydrogels with mechanical properties similar to human soft tissues. *J. Mater. Chem. B* **7**, 3512-3519 (2019).

[4] Laurano, R. et al. Dual stimuli-responsive polyurethane-based hydrogels as smart drug delivery carriers for the advanced treatment of chronic skin wounds. *Bioact. Mater.* **6**, 3013-3024 (2021).

[5] Wendels, S. & Avérous, L. Biobased polyurethanes for biomedical applications. *Bioact. Mater.* **6**, 1083-1106 (2021).

Revision: For avoiding misunderstanding, only the scaffolds accomplished water-responsive deformation were referred to as “hydrogel scaffolds” in the revised manuscript. *In Page 3 Line 90 (Introduction):* “After immersion in 37 °C water, the 1D roll recovers to the initial 2D pattern and absorbs water to achieve programmed deformation, **finally forming a 3D hydrogel scaffold.**”

The morphology and weight of the printed pattern before and after swelling are compared in following Figure 1 (not listed in the revised manuscript). The dry butterfly printed with DTPU-0.5-4k was 383.4 mg, while the hydrated hydrogel butterfly weighed 827.1 mg. The yellow color of the hydrogel is a result of the color of the chain extender DA-diol. Compared to dry butterfly, the hydrated hydrogel butterfly

underwent some degree of volume expansion.

Figure 1. Comparison of the morphology and weight of the printed butterfly before and after swelling.

Q3. PUHI was synthesized in DMF. Is the synthesized prepolymer solution soluble in water? If not, then how could it be the hydrogel ink? If yes, then what is the water percentage to make PU hydrogel ink, and how would it be possible to print that ink at 130°C?

Response: Thanks for this comment; our description of hydrogel ink is indeed incorrect. The synthesis of the dynamic thermoset polyurethane (DTPU) was divided into two steps: prepolymerization and chain extension, and both were carried out in a mixture of DMF and THF solvents. After this, the reaction solution was poured into the Teflon mold and the solvents were completely evaporated by post-processing. Therefore, it's truth that this is not hydrogel ink in this state, and we changed the “hydrogel ink” to “dynamic thermoset polyurethane”. The material used for printing is the dry DTPU after the two-step reaction and post-processing, which can only swell but not dissolve in water.

Revision: We have changed the “polyurethane-based hydrogel ink” and “polyurethane-based hydrogel inks” to “dynamic thermoset polyurethane” and “dynamic thermoset polyurethanes”, respectively. All the “PUHI” and “PUHIs” have been changed to “DTPU” and “DTPUs”, respectively. And only the scaffolds that accomplished water-

responsive deformation were referred to as “hydrogel scaffolds” in the revised manuscript. *In Page 3 Line 90 (Introduction)*: “After immersion in 37 °C water, the 1D roll recovers to the initial 2D pattern and absorbs water to achieve programmed deformation, **finally forming a 3D hydrogel scaffold.**”

Q4. Supplementary Fig 9 C, the elongation and fracture strength of the pristine tensile stress-strain curve is less than the healing curve of 12 h, which means that healing efficiency is more than 100%. Can the author explain why elongation and fracture strength are more for the healed sample than the original?

Response: We appreciate the reviewer for pointing it out. The self-healing properties of the prepared DTPU-0.5-4k were tested with three sets of parallel samples separately for each healing condition. The healing efficiency was calculated to be up to $94.8 \pm 7.3\%$ for 12 h of healing at 37 °C. And the added statistical analysis showed that the strength of the samples under this healing condition was not significantly different from that of the pristine samples, indicating that the sample incision was almost completely repaired. In that case, it may happen that the elongation and fracture strength of a single repaired sample are higher than those of a single pristine sample, since the tensile properties of the pristine DTPU-0.5-4k itself fluctuate within a certain range, but we should choose the more representative data curve.

Revision: In the modified **Supplementary Figure 9**, we added the data of tensile strength (**Supplementary Fig. 9b, e**), and choose the more representative data curve in **Supplementary Figure 9d**. We added a description of the significant difference analysis. *In Page 5 Line 136 (Results)*: “After 12 h of healing at 37 °C, **the tensile strengths of the healed samples were not significantly different from those of the pristine samples, corresponding to the high healing efficiency of 94.8%.**” New **Supplementary Figure 9** has been added to the revised manuscript.

Supplementary Fig. 9 Tensile stress strain curves (a), tensile strength (b), and healing efficiency (c) of the self-healed DTPU-0.5-4k at various healing temperatures for 1 h. Tensile stress strain curves (d), tensile strength (e), and healing efficiency (f) of the self-healed DTPU-0.5-4k at 37 °C for various healing time. Values in (b), (c), (e), and (f) represent mean \pm SD. One-way analysis of variance (ANOVA), Tukey's post hoc test. ($n=3$ independent samples. $***P < 0.001$, $****P < 0.0001$, ns: no significance).

Q5. Were all the studies for DMA, Mechanical, swelling, shape memory performed for PUHI i.e, hydrogel ink? Does the printed scaffold also show the same property? Since the final application is based on the printed scaffold, authors should study the above-mentioned properties of printed scaffolds, not the hydrogel ink. There is no value in studying the ink.

Response: We thank the reviewer for this suggestion. In Figure 5d and e (the first draft), we only investigated the effect of the print path on the mechanical properties. The results showed that printing in an orthogonal way avoids the negative effects of the printing process on the mechanical properties. In the revised manuscript, we have comprehensively added the shape memory function, mechanical properties, swelling behaviors, and thermodynamic properties of the printed scaffolds.

Revision: In Page 11 Line 303 (**Results**): “To comprehensively examine the effect of printing procedures on the above-mentioned thermal-induced shape memory, swelling-stiffening behavior, and water-triggered deformation, DTPU-0.25-4k and DTPU-0.5-

4k were used to print scaffolds through an orthogonal path with different line distance. The dry scaffolds with 0.4 mm line distance showed mechanical properties comparable to those of the cast molded samples (Fig. 5, e-g). However, the mechanical properties of the dry scaffolds printed with 1 mm line distance were impaired. For example, dry DTPU-0.5-4k scaffold printed with 1 mm line distance showed σ_b of 0.9 MPa, E of 0.3 MPa, and ε_b of 294.0%, respectively, significantly lower than those of the dry cast DTPU-0.5-4k samples (σ_b of 3.0 MPa, E of 0.7 MPa, and ε_b of 822.3%). The mechanical deterioration is attributed to the pore structure within the printed scaffold, reducing the volume of material dissipating energy. In contrast, no pore existed in the scaffold with 0.4 mm line distance due to the thick line diameter (Supplementary Fig. 20). The ε_b of the hydrated scaffolds decreased significantly, indicating that the print defects were amplified during the swelling of printed samples (Supplementary Fig. 23). While the maintenance of σ_b and E was attributed to the absence of pore in samples printed at 0.4 mm line distance. We assumed that the retention of E for hydrated scaffolds printed with 1 mm line distance is due to swelling resulting in a smaller pore structure, thereby increasing the material volume for dissipating energy in the stretching direction. In short, the printing parameter of line distance has an impact on the mechanical properties of scaffolds due to the presence of the pore structure. However, the E of hydrated scaffolds were significantly higher than those of the corresponding dry scaffolds, indicating that the printing procedure does not affect the swelling-stiffening behavior.

The storage modulus of the printed scaffolds decreased with the increase in printing line distance, which is consistent with the influence of line distance on mechanics (Supplementary Fig. 24a). However, the peak of $\tan \delta$ in the low-temperature region disappeared (Supplementary Fig. 24b), which is because the rapid temperature drop after extrusion printing results in insufficient time to form an obvious phase separation structure. After water absorption, the hydrated scaffolds exhibited $\tan \delta$ peaks at -50 °C (Supplementary Fig. 24c), indicating that swelling led to microphase separation formed between hydrophilic and hydrophobic chain segments. Based on the sharp change in the storage modulus of the scaffolds near body temperature, the printed butterfly and

crab were fixed into a vivid temporary shape at 4 °C. In a 37 °C environment, the 3D temporary shape quickly reverted to the original flat pattern (Supplementary Fig. 25), demonstrating that the body temperature-triggered shape memory function of the printed structure is not affected. Finally, the influence of the printing process on the swelling ratio was determined. Benefiting from the pore structures [45], the printed scaffolds displayed shorter balancing time within 5 min (Supplementary Fig. 26). While the final swelling ratios were basically consistent with those of cast molded samples (Fig. 5h), indicating that the printing process has no effect on the swelling ratio of DTPU. Therefore, it is feasible to construct swelling mismatch structures by multi-material printing.” The mechanical properties of printed DTPU have been added to the manuscript Figure 5e-g and Supplementary Figure 23. The swelling behaviors of printed DTPU have been added to the manuscript Figure 5h and Supplementary Figure 26. The DMA of printed DTPU has been added to the manuscript Supplementary Figure 24. The shape memory of printed DTPU has been added to the manuscript Supplementary Figure 25.

Fig. 5. Printability of DTPU. (a) Oscillatory temperature sweep of DTPU showing transition temperature between elastic state and viscoelastic liquid state. (b) Viscosity of DTPUs as a function of temperature. (c) Oscillatory alternating temperature sweep of DTPU-0.25-4k. (d) Printing paths and as-prepared structures printed using DTPU-0.25-4k. Tensile strength (e), Young's modulus (f), and tensile strain at break (g) of dry scaffolds constructed with different molding methods. (h) Swelling ratio of scaffolds constructed with different molding methods. Values in (e), (f), (g), and (h) represent mean \pm SD. One-way analysis of variance (ANOVA), Tukey's post hoc test. ($n = 3$ independent samples. $*P < 0.05$, $**P < 0.01$, $***P < 0.001$, ns: no significance).

Supplementary Fig. 23 Tensile strength (a), Young's modulus (b), and tensile strain at break (c) of hydrated scaffolds constructed with different molding methods. Values represent mean \pm SD. One-way analysis of variance (ANOVA), Tukey's post hoc test. ($n = 3$ independent samples. $*P < 0.05$, $**P < 0.01$, $***P < 0.001$, ns: no significance).

Supplementary Fig. 24 Storage modulus-temperature curve (a), and $\tan \delta$ -temperature curve (b) of dry flat scaffolds printed using DTPU-0.25-4k or DTPU-0.5-4k with different line distance. $\tan \delta$ -temperature curve of hydrated DTPU-0.25-4k scaffolds printed with different line distance (c).

Supplementary Fig. 25 Shape recovery of a crab (a) and a butterfly (b) printed with DTPU-0.25-4k from temporary vivid 3D shapes to initial 2D shapes. Scale bars:10 mm.

[45] Kuang, X. et al. Advances in 4D printing: Materials and applications. *Adv. Funct. Mater.* **29**, 1805290 (2019).

Q6. What is the molar ratio between NCO and di-ol? As per the reaction scheme, NCO functional grp will remain in the PHUI system, then how do the authors avoid the toxicity of the isocyanate group?

Response: This is an important question for the *in vivo* application of printed scaffold. In this work, the -NCO was provided by IPDI, while the -OH was supplied by PCL-triol, PEG, and DA-diol. The isocyanate index R, determined by the molar ratio of isocyanate group to hydroxide group, was 1.06, indicating that the system still contains unreacted isocyanate groups after the chain extension reaction. In order to eliminate the consumption of the isocyanate groups by traces of water in the monomer and solvent, we set R slightly larger than 1. Although toxic -NCO may exist in the reaction solution obtained after the two-step reaction, it could react with water in the air during the post-processing [1,2].

The FTIR in Supplementary Figure 2 showed that the dry dynamic thermoset polyurethane (DTPU) did not have -N=C=O stretching vibration peaks in the wavenumber range of 2200-2280 cm⁻¹, indicating that the excess -NCO groups inside the DTPU were completely consumed at this time. In addition, the final printed scaffolds were soaked in PBS in vitro in order to determine their programmable deformability prior to actual application. These manipulations also ensure complete depletion of the -NCO. We also supplemented the cytotoxicity assay and subcutaneous implantation assay of prepared polyurethanes and printed scaffolds for evaluating their biocompatibility.

[1] Grzęda, D. et al. Cytotoxic properties of polyurethane foams for biomedical applications as a function of isocyanate index. *Polymers* **15**, 2754 (2023).

[2] Cao, H., Li, B., Jiang, X., Zhu, X. & Kong, X. Z. Fluorescent linear polyurea based on toluene diisocyanate: Easy preparation, broad emission and potential applications. *Chem. Eng. J.* **399**, 125867 (2020).

Revision: We added the cytotoxicity assay and subcutaneous implantation experiments of prepared polyurethanes and printed scaffolds according to ISO 10993-12:2021, ISO 10993-6:2016, and ISO 10993-5:2009. As shown in **Figure 7d** (revised revision), the cytotoxicity tests confirmed that all the DTPUs with different components were non-toxic to L929 cells. And the printing process had no significant effect on cytotoxicity. Histological evaluation also showed that the inflammatory response decreased as implantation duration was increased (**Fig. 7, e and g**). *In Page 4 Line 111 (Results):* “The successful synthesis of **DTPU** was verified through Fourier-transform infrared spectroscopy (FTIR) (Supplementary Fig. 2). **The absence of characteristic absorption at 2200-2280 cm⁻¹ demonstrated complete depletion of the excess isocyanate groups.**” *In Page 14 Line 400 (Results):* “Moreover, cytotoxicity tests confirmed that all the **DTPU films** were non-toxic to L929 cells (**Supplementary Fig. 28**). **And the scaffolds printed with DTPU-0.25-4k or DTPU-0.5-4k also demonstrated excellent biocompatibility. To investigate the foreign body response, scaffolds printed with DTPU-0.25-4k or DTPU-0.5-4k were subcutaneously implanted in SD rats. On week 2**

and week 4, the scaffolds were completely covered by the surrounding tissue, and no obvious difference around the tissue could be seen (Fig. 7d). Histological evaluation through hematoxylin-eosin staining also showed a postoperative inflammatory response with accumulations of inflammatory cells in sham-control and experimental groups after 1 week (Fig. 7e). But the inflammatory response gradually disappeared at 2 and 4 week, indicating good *in vivo* biocompatibility of the DTPU scaffolds. Therefore, the developed DTPUs provided a strategy for minimally invasive delivery of void-filling scaffolds, showing potential in biomedical applications.” *In Page 15 Line 446 (Methods):* “To eliminate the consumption of the isocyanate groups by traces of water in the monomer and solvent, the molar ratio of the isocyanate group to hydroxyl (-NCO/-OH) was 1.06. The excess -NCO inside the DTPU were completely depleted during the post-processing.” New Figure 7 and Supplementary Figure 28 have been added to the manuscript.

Fig. 7. Transcatheter delivery and biocompatibility. (a) Schematic of 4D printed scaffold for transcatheter delivery and subcutaneous multi-dimensional deformation. (b) Digital images revealing the transcatheter delivery procedure and multi-endogenous stimuli responsiveness of the 4D printed structure. (c) Photographs showing the feasibility of 4D printed scaffolds to complete 1D to 3D deformation subcutaneously in SD rats. (d) Gross pictures of the implanted scaffolds and (e) Histological evaluation through hematoxylin-eosin staining.

surrounding tissues at different timepoints. (e) Histological evaluation of the printed DTPU-0.25-4k and DTPU-0.5-4k scaffolds at different timepoints. Scale bars in (b) and (d) were 10 mm, scale bars in (e) were 1 mm.

Supplementary Fig. 28 Cytotoxicity analysis of DTPUs casted or printed with different PEG/PCL-triol and M_{PEG} . Values represent mean \pm SD. One-way analysis of variance (ANOVA), Tukey's post hoc test. (n = 3 independent samples. **** $P < 0.0001$).

Q7. In figure 2 f, why the loss modulus increased near -20°C and decreased at high temperatures is unclear. Can the author explain these phenomena?

Response: The temperature corresponding to the peak of the loss modulus represents the glass transition temperature (T_g) of the polymer [1,2]. In this work, the peak of loss modulus around -20°C represents the T_g of the polyurethane soft segments. Below this temperature, the polyurethane molecular chain movement freezes. And the deformation is mainly realized by changes in chemical bond length and bond angle, at which time the storage modulus is high. As the temperature rises to near the T_g of the soft segments, the thawed soft segments begin to move, resulting in a gradual increase in internal friction and system viscosity. The storage modulus of the polyurethane starts to decrease, while the loss modulus increases. The peak of the loss modulus is attributed to the maximum dissipation of mechanical energy, indicating a transition of the

dominant motion unit in that temperature region [3]. As the temperature continues to increase, the polyurethane reaches a high-elastic state and becomes soft, so the loss modulus decreases. The loss modulus rises again until the temperature rises to near the T_g of the polyurethane hard segments. Continuing increasing the temperature, the retro-DA reaction leads to a gradual dissociation of the dynamic crosslinked network. Both the storage modulus and loss modulus decline continuously. It could be seen that the T_g of the soft segments and the T_g of the hard segments decreased with the increase of M_{PEG} and PEG/PCL-triol, which corresponds to the effect of components on mechanics and structure of the dynamic thermoset polyurethane.

[1] Nam, J. & Seferis, J. C. Viscoelastic characterization of phenolic resin-carbon fiber composite degradation process. *J. Polym. Sci. Pol. Phys.* **37**, 907-918 (1999).

[2] Kumar, R., Varshney, S., Kar, K. K. & Dasgupta, K. Enhanced thermo-mechanical and electrical properties of carbon-carbon composites using human hair derived carbon powder as reinforcing filler. *Adv. Powder Technol.* **29**, 1417-1432 (2018).

[3] Kumar, R., Kar, K. K. & Dasgupta, K. Enhanced electrical, mechanical, and viscoelastic properties of carbon-carbon composites using carbon nanotubes coated carbon textile as reinforcement. *J. Compos. Mater.* **55**, 1733-1748 (2021).

Revision: In the revised manuscript, we added an explanation of the significance of the loss modulus peak. *In Page 6 Line 174 (Results):* “Notably, the T_g of the soft segments could be obtained from the loss modulus peaks in DMA, **which represent a transition of the dominant motion unit in that temperature region [36, 37].**”

[36] Kumar, R., Varshney, S., Kar, K. K. & Dasgupta, K. Enhanced thermo-mechanical and electrical properties of carbon-carbon composites using human hair derived carbon powder as reinforcing filler. *Adv. Powder Technol.* **29**, 1417-1432 (2018).

[37] Kumar, R., Kar, K. K. & Dasgupta, K. Enhanced electrical, mechanical, and viscoelastic properties of carbon-carbon composites using carbon nanotubes coated carbon textile as reinforcement. *J. Compos. Mater.* **55**, 1733-1748 (2021).

Q8. Since the authors are using extrusion-based printing, in Figure 2, the plot of viscosity with temperature and the thixotropy test at various temperatures would be more appropriate than storage and loss modulus plots.

Response: We thank the reviewer for this suggestion. First, please allow me to explain

why DMA test is necessary. In this work, we constructed a new DTPU system using several common monomers. Therefore, we spent more pages in the first half of the article to characterize the various properties such as swelling-stiffening, body temperature-triggered shape memory, and water-triggered deformation, and finally applied the DTPU to the construction of 4D printed scaffolds. In Figure 2, the theme was to clarify the effect of M_{PEG} and PEG/PCL-triol on the mechanical properties of the fabricated polyurethanes. Dynamic mechanical analysis was tested to confirm the changes in the microstructure of different polyurethanes.

In the first draft, the crossover of G' and G'' , representing the transition temperature of the DTPU from solid to fluid state, had been evaluated to demonstrate the printability (Fig. 5a). And the change in viscosity with temperature was measured to investigate extrudability (Fig. 5b).

Revision: In the revised manuscript, we added temperature thixotropy results of DTPU-0.25-4k and DTPU-0.5-4k in Figure 5c and Supplementary Figure 21. The DTPU showed a rapidly reversible solid \leftrightarrow fluid transition behavior when subjected to alternative low temperature (30 °C) and high temperature (130 °C) treatment, indicating repeatable printing capability. We hope that the complementary rheological experiments will effectively prove the printability of the DTPU through extrusion-based fused deposition modeling (FDM). *In Page 10 Line 286 (Results):* “Considering similar viscosity and shape fidelity, DTPU-0.25-4k and DTPU-0.5-4k were selected for multi-material 4D printing. To further test the temperature thixotropy, DTPU-0.25-4k and DTPU-0.5-4k were subjected to alternating low (30 °C) and high (130 °C) temperatures. Rapid and reversible transitions from elastic state to viscous liquid state were observed, indicating repeatable printing capability of DTPU (Fig. 5c and Supplementary Fig. 21).” New Figure 5 and Supplementary Figure 21 have been added to the manuscript.

Fig. 5. Printability of DTPUs. (a) Oscillatory temperature sweep of DTPU showing transition temperature between elastic state and viscoelastic liquid state. (b) Viscosity of DTPUs as a function of temperature. (c) Oscillatory alternating temperature sweep of DTPU-0.25-4k. (d) Printing paths and as-prepared structures printed using DTPU-0.25-4k. Tensile strength (e), Young's modulus (f), and tensile strain at break (g) of dry scaffolds constructed with different molding methods. (h) Swelling ratio of scaffolds constructed with different molding methods. Values in (e), (f), (g), and (h) represent mean \pm SD. One-way analysis of variance (ANOVA), Tukey's post hoc test. ($n = 3$ independent samples. $*P < 0.05$, $**P < 0.01$, $***P < 0.001$, ns: no significance).

Supplementary Figure 21. Oscillatory alternating temperature sweep of DTPU-0.5-4k.

Q9. What happens if a deformed or fixed structure is exposed to temperature and water simultaneously, and what would the recovery and final shape programmability be? Then, will recovery and programmability of the final shape be affected? Autho should study this.

Response: As shown in Figure 6f and Supplementary Movie 3, the fixed 1D rod unfolded and morphed to a preprogrammed 3D structure within 1 min when delivered to 37 °C water, which means exposure to temperature and water simultaneously. The deformation process shows that body temperature-triggered shape recovery and water-responsive programmable deformation are simultaneous, while the temperature response is slightly faster. However, curling and fixing shape in only one direction does not fully demonstrate the effect of low-temperature fixation on the final shape.

In the revised manuscript, we have tried our best to measure the deformations of the printed scaffold fixed in different ways when exposed to 37 °C water and compared them with the water-responsive programmable deformation. As shown in Supplementary Figure 27, the printed 2D pattern bent approximately 360° to form a closed cylinder when placed directly in 37 °C water (the blue dotted arrow). First, the 2D pattern was rolled and fixed into a 1D short rod around the short axis inwards (DTPU-0.25-4k layer facing inside) or outwards (DTPU-0.5-4k layer facing inside),

respectively. After immersed in 37 °C water, temporary shapes morphed into 3D cylinders within 1 min, indicating that fixation by curling around the short axis does not affect the final shape. Then, the 2D pattern was rolled and fixed into a 1D long rod around the long axis inwards or outwards, respectively. After immersed in 37 °C water, temporary shapes morphed into closed cylinders within 1 min. However, misalignment was found at the cylinder closure, suggesting that fixation by curling around the long axis affects the final shape. We hypothesize that this is due to the excessively high water-response rate, as this type of fixing causes an uneven rate of water absorption between inside and outside, which leads to tilting when bending. It is worth noting that due to the porous nature of the printed structure, this type of fixation does not have a significant impact on the final shape. Finally, the 2D pattern was folded inward or outward from the center along the short axis and fixed. After placed in 37 °C water, temporary shapes morphed into 3D cylinders within 1 min, implying that fixation by folding along the short axis does not affect the final shape. In summary, the pre-determined 3D shape of the printed scaffold is largely independent from the low-temperature fixation type, so that the final shape for *in vivo* application is overall controllable.

Revision: *In Page 13 Line 379 (Results):* “Further, the printed scaffolds were fixed in different ways to examine the effect of simultaneous temperature and water responses on the final shape (Supplementary Fig. 27). Whether inwards (DTPU-0.25-4k layer facing inside) or outwards (DTPU-0.5-4k layer facing inside), fixing by folding or curling the 2D pattern around the short axis did not affect the final pre-determined 3D shape. However, slight misalignment was found in the final 3D shape when fixing by curling around the long axis. It is highly plausible due to the excessively high water-response rate, as this type of fixing causes an uneven rate of water absorption between inside and outside, which leads to tilting when bending. Attributed to the porous nature of the printed structure, the slight misalignment showed little impact on the final 3D shape.” New Supplementary Figure 27 has been added to the manuscript.

Supplementary Fig. 27 Images of final 3D shapes formed in response to simultaneous temperature and water under different fixation patterns. The inner layer shows the initial shape of the 2D flat pattern and the corresponding model. The middle layer exhibits the temporary shape after being rolled or folded in different ways and then fixed at 4 °C. The corresponding model demonstrates the specific fixation method. The outer layer shows the final shape of the corresponding temporary shape in response to simultaneous stimulation of temperature and water (37 °C water).

Response to Reviewer #2:

Comments: This is an excellent manuscript dealing with memory shapes polymers how 4d printing properties. The manuscript is very well written. The mechanical properties are well supported by AFM, chemical analysis and structural properties. The visual material is very useful.

Response: We sincerely appreciate your positive evaluation of our manuscript. It is truly encouraging to receive your commendation for the excellence of our work in exploring shape memory polymers through 4D printing.

Q1. No statistics for the measurements are provided in the case mechanical characterization and healing efficiency. Can the authors comment on that and demonstrate that the measurements are reproducible?

Response: We thank the reviewer for this suggestion. In the revised manuscript, we added the statistical analysis for the mechanical properties and healing efficiency of the prepared polyurethanes. And we stated in the **Methods** section that the relevant measurements were conducted using three independent samples to ensure the reproducibility. To demonstrate the swelling-stiffening properties, significant differences in each mechanical property of the polyurethanes before and after swelling were determined by two-tailed *t*-test (**Supplementary Fig. 10**). To compare the self-healing efficiency under different healing conditions, one-way analysis of variance (ANOVA) with Tukey post-test was carried out (**Supplementary Fig. 9**). We also added a discussion of significant differences in the corresponding section (please see the revised text).

Revision: *In Page 21 Line 625 (Methods):* “**Statistics and reproducibility: Statistical analysis was performed with GraphPad Prism9.5. Statistical significance between two groups was calculated using two-tailed *t*-test. Multiple comparisons were assessed using one-way analysis of variance (ANOVA) with Tukey post-test. Three independent experiments were carried out and data were expressed as mean \pm standard deviation.**”
New **Supplementary Figure 9** and **Supplementary Figure 10** have been added to the manuscript.

Supplementary Fig. 10 Tensile strength (a), tensile strain at break (b), Young's modulus (c), and toughness (d) of dry and hydrated DTPU with different PEG/PCL-triol and M_{PEG} . Values represent mean \pm SD. Statistical analysis: two-tailed t -test. ($n = 3$ independent samples. * $P < 0.05$, ** $P < 0.01$, *** $P < 0.001$, **** $P < 0.0001$, ns: no significance).

Supplementary Fig. 9 Tensile stress strain curves (a), tensile strength (b), and healing efficiency (c) of the self-healed DTPU-0.5-4k at various heating temperatures for 1 h. Tensile stress strain curves (d), tensile strength (e), and healing efficiency (f) of the self-healed DTPU-0.5-4k at 37 °C for various heating time. Values in (b), (c), (e), and (f) represent mean \pm SD. One-way analysis of variance (ANOVA), Tukey's post hoc test. ($n = 3$ independent samples. *** $P < 0.001$, **** $P < 0.0001$, ns: no significance).

Q2. The text under the AFM images and the scale bar are barely visible. Also, the quality of the images probably needs to be improved.

Response: Thanks for this comment. In the revised manuscript, we removed the original fuzzy scale of the AFM data and added a clearer scale. The quality of the AFM images has also been updated. The modified **Figure 2d** is shown as following.

Fig. 2d Typical AFM images of different **DTPU** films. Size $1 \mu\text{m} \times 1 \mu\text{m}$.

Q3. For self healing please provide more details on how the cut was performed.

Response: We thank the reviewer for this suggestion. In the revised manuscript, specific details about the self-healing process were added to the **Mechanical testing** section in **Methods**.

Revision: *In Page 16 Line 473 (Methods):* “For the “scratch-healing” test, the prepared DTPU film ($30 \text{ mm} \times 10 \text{ mm} \times 0.5 \text{ mm}$) was attached to a slide by double-sided tape. And a scratch was created at the center of the film by using a scalpel. Subsequently, the slide carrying the film was placed in $60 \text{ }^\circ\text{C}$ oven for varying periods of time to examine the self-healing. To quantify the healing efficiency, the prepared DTPU films were cut into dumbbell-shaped specimens using a IV dumbbell cutter that conformed to the GB/T 528-2009 standard. And the dumbbell-shaped specimens were cut in half with a blade. The two parts were immediately reattached and placed under different conditions for healing. No external force was applied to the specimens during the entire healing process.”

Response to Reviewer #3:

Comments: This paper characterized the development of novel shape memory polymers for use with 3D printing applications, for the generation of implants that would require minimally invasive surgeries. The work presented is novel, and is of interest to the scientific community. The authors did a wonderful job fully characterizing their materials - as far as mechanical, swelling, and thermal-induced shape memory. I recommend this paper for publication following minor revisions:

Response: We sincerely appreciate your thoughtful review of our manuscript and are delighted to learn that you find our work to be novel and of interest to the scientific community. Your constructive feedback is invaluable to us, and we are committed to addressing your suggestions to further enhance the quality of our manuscript.

Q1. Since the paper is built around potential *in vivo* applications I think the paper fell a little short on cytocompatibility assays, the authors did evaluate toxicity generated by leachables following a 2 day incubation of their material in media - I feel a better evaluate would be to follow: ISO 10993-12:2021(E) (Biological evaluation of medical devices — Part 12: Sample preparation and reference materials). While *in vivo* data may be beyond the scope of this communication, that would also have been satisfying to see what kind of foreign body response the material induces.

Response: We appreciate the reviewer's constructive suggestion. In the revised manuscript, cytotoxicity assay was retested to follow ISO 10993-12:2021(E), and ISO 10993-5:2009. For long-term implanted materials, the leaching condition was changed to 72 h at 37 °C. The hydrogels equilibrated in sterile PBS were subjected to half an hour of UV irradiation prior to the extraction operation. Cells cultured in extracts without material samples, extracts of HD-polyethylene (HDPE), and phenol solution, were considered as blank, negative, and positive controls, respectively. The positive control group (phenol solution) had significantly lower cell viability than the blank

control group. While the cell viabilities of the negative control group (HDPE) and the experimental groups were not significantly different from that of the blank control group (Supplementary Fig. 28), indicating good biocompatibility. Meanwhile, the printed DTPU-0.25-4k and printed DTPU-0.5-4k groups demonstrated that the printing process had no effect on the biocompatibility. Histological evaluation also showed that the inflammatory response decreased as implantation duration was increased (Fig. 7, d and e).

Revision: *In Page 14 Line 400 (Results):* “Moreover, cytotoxicity tests confirmed that all the DTPU films were non-toxic to L929 cells (Supplementary Fig. 28). And the scaffolds printed with DTPU-0.25-4k or DTPU-0.5-4k also demonstrated excellent biocompatibility. To investigate the foreign body response, scaffolds printed with DTPU-0.25-4k or DTPU-0.5-4k were subcutaneously implanted in SD rats. On week 2 and week 4, the scaffolds were completely covered by the surrounding tissue, and no obvious difference around the tissue could be seen (Fig. 7d). Histological evaluation through hematoxylin-eosin staining also showed a postoperative inflammatory response with accumulations of inflammatory cells in sham-control and experimental groups after 1 week (Fig. 7e). But the inflammatory response gradually disappeared at 2 and 4 week, indicating good *in vivo* biocompatibility of the DTPU scaffolds. Therefore, the developed DTPUs provided a strategy for minimally invasive delivery of void-filling scaffolds, showing potential in biomedical applications.” *In Page 20 Line 587 (Methods):* “The cast DTPU films and the printed 2D scaffolds were equilibrated in PBS and sterilized using ultraviolet for 0.5 h. And sterilized DTPU hydrogels were immersed in Roswell Park Memorial Institute medium 1640 (RPMI-1640) for 72 h at 37 °C to prepare the leachates. The volume of RPMI-1640 equaled 1 mL per 0.1 g of the equilibrated DTPU. Mouse fibroblast (L929) cells were seeded in 96-well plates at a density of 2×10^4 cells/well and incubated for 24 h at 37 °C in a 5% CO₂ humidified atmosphere. Then, the culture medium was replaced with 100 μL complete growth medium and 100 μL leachate. After 24 h incubation, the medium was removed and 100 μL medium containing MTT (0.5 mg/mL) was added into the wells and the incubation

was continued for 4 h. The medium was removed and 150 μ L dimethyl sulfoxide was added to dissolve formed formazan crystals. After shaking for 10 min, the spectrophotometric readings were operated at 490 nm by a Multiskan GO plate reader (Thermo Fisher Scientific, USA). Cells cultured in extracts without material samples, extracts of HD-polyethylene, and phenol solution, were considered as blank, negative, and positive controls, respectively. Cell viability was calculated as relative percentage of living cells compared to the blank control [47].” New Figure 7 and Supplementary Figure 28 have been added to the manuscript.

Fig. 7. Transcatheter delivery and biocompatibility. (a) Schematic of 4D printed scaffold for transcatheter delivery and subcutaneous multi-dimensional deformation. (b) Digital images revealing the transcatheter delivery procedure and multi-endogenous stimuli responsiveness of the 4D printed structure. (c) Photographs showing the feasibility of 4D printed scaffolds to complete 1D to 3D deformation subcutaneously in SD rats. (d) Gross pictures of the implanted scaffolds and surrounding tissues at different timepoints. (e) Histological evaluation of the printed DTPU-0.25-4k and DTPU-0.5-4k scaffolds at different timepoints. Scale bars in (b) and (d) were 10 mm, scale bars in (e) were 1 mm.

Supplementary Fig. 28 Cytotoxicity analysis of DTPUs casted or printed with different PEG/PCL-triol and M_{PEG} . Values represent mean \pm SD. One-way analysis of variance (ANOVA), Tukey's post hoc test. (n = 3 independent samples. **** $P < 0.0001$).

Q2. Figure 2 a - replace "cutted" with cut,

Response: Thanks for this suggestion. We have replaced the "cutted" in Figure 2a with "cut".

Fig. 2a Demonstration of self-healing between fractured surfaces through the reversible DA reaction.

Q3. Figure 3a - please include an image of the printed part before manipulation

Response: In the revised manuscript, we added the initial morphology of the dry and hydrated samples used to demonstrate the swelling-stiffening properties, respectively. The modified **Figure 3a** is shown as following.

Fig. 3a Digital photos showing the reversibility of swelling-stiffening behavior of **DTPU-0.25-4k**.

Q4. Figure 3c - could you present the actual AFM data? What is the scale?

Response: We thank the reviewer for this suggestion. In the revised manuscript, we added a clearer scale to the AFM data. The quality of the AFM images has also been updated. The modified **Figure 2d** is shown as following. The description of Figure 3c was added. *In Page 7 Line 206 (Results):* “Upon water absorption, hydrophilic PEG segments aggregated and separated from the hydrophobic segments because of the incompatibility of the two segments (**Fig. 3c**).”

Fig. 2d Typical AFM images of different **DTPU** films. Size $1\ \mu\text{m} \times 1\ \mu\text{m}$.

Q5. Figure 4b include an image of the original part without weight manipulation

Response: In the revised manuscript, we added images of the initial shape of the

support scaffold from different perspectives. The modified Figure 4b is shown as following.

Fig. 4b Digital photos showing the capacity of DTPU-0.25-4k to resist deformation at 4 °C and 37 °C, respectively.

Q6. For the *in vivo* experiment - why was the 37°C water flush necessary? Could the researchers just waited longer or have placed a heating pad over the animal?

Response: We appreciate the reviewer’s constructive suggestion. In this work, the transition from 1D to 3D of the printed scaffold is best realized by undergoing body temperature stimulation and body fluid stimulation successively, without any impact on the final shape. In addition, the need for sufficient space during the morphing from 1D to 3D is a current problem with our scaffold. If the scaffold is implanted *in vivo* without additional 37 °C water immersion, the water response rate of the portion of the scaffold in contact with the body surface will be higher than the temperature response rate of the entire scaffold, which may affect the final 3D shape of the scaffold. Furthermore, the portion of the scaffold not contacting the body surface absorbs water at a lower rate, which prolongs the response time for completed deformation. Using a heat lamp to raise the ambient temperature to 37 °C, the scaffold will be difficult to deliver transcatheter due to premature temperature-sensitive shape recovery. In the future work, we hope to be able to control the start time of different responses, as reported by Tao Xie [1], as a way to optimize the surgical procedure.

[1] Ni, C. et al. Shape memory polymer with programmable recovery onset. *Nature* **622**, 748-753 (2023).

Revision: *In Page 14 Line 394 (Results):* “Water at 37 °C was injected under the skin

of Sprague Dawley (SD) rats to simulate body fluids,” has been modified as “Water at 37 °C was injected under the skin of Sprague Dawley (SD) rats to **create deformation space and allow the scaffold to swelling homogeneously,**”

Q7. Do each of the shapes have a specific time scale for when they fully transform at 37°C?

Response: For the cast models, the temperature response and water response times are roughly 5-10 min, based on the thickness of the models (1-2 mm). For the print scaffolds, the temperature response and water response times are within 1-10 min due to the thin and porous print structures. It is worth mentioning that the fast response time scale within 10 min is only available for DTPU-0.5-2k, DTPU-0.25-4k, and DTPU-0.5-4k. The water response times for DTPU-0.25-1k, DTPU-0.5-1k, and DTPU-0.25-2k are at least 4 h.

Revision: In the revised manuscript, we have added a description of the response time of the printed scaffolds. *In Page 13 Line 376 (Results):* “**It is worth mentioning that all printed scaffolds can complete body temperature-triggered shape recovery and water-responsive programmable deformation within 10 min.**”

REVIEWERS' COMMENTS

Reviewer #1 (Remarks to the Author):

The authors have revised the manuscript to answer the technical concerns. While the work is technically sound, I still have my concerns with the extent of novelty.

Reviewer #2 (Remarks to the Author):

I am satisfied with the revisions

Reviewer #3 (Remarks to the Author):

The authors have addressed all questions in a robust manner, and have updated the manuscript accordingly. I recommend publication of this revised manuscript